# RASSF1A controls tissue stiffness and cancer stem-like cells in lung adenocarcinoma

Daniela Pankova[1], Yanyan Jiang[1,2], Maria Chatzifrangkeskou[1] (ID), Iolanda Vendrell[1,3], Jon Buzzelli[1], Anderson Ryan[1,2], Cameron Brown[4] & Eric O'Neill[1,5,*] (ID)

## Abstract

Lung cancer remains the leading cause of cancer-related death due to poor treatment responses and resistance arising from tumour heterogeneity. Here, we show that adverse prognosis associated with epigenetic silencing of the tumour suppressor RASSF1A is due to increased deposition of extracellular matrix (ECM), tumour stiffness and metastatic dissemination *in vitro* and *in vivo*. We find that lung cancer cells with RASSF1A promoter methylation display constitutive nuclear YAP1 accumulation and expression of prolyl 4-hydroxylase alpha-2 (P4HA2) which increases collagen deposition. Furthermore, we identify that elevated collagen creates a stiff ECM which in turn triggers cancer stem-like programming and metastatic dissemination *in vivo*. Re-expression of RASSF1A or inhibition of P4HA2 activity reverses these effects and increases markers of lung differentiation (TTF-1 and Mucin 5B). Our study identifies RASSF1A as a clinical biomarker associated with mechanical properties of ECM which increases the levels of cancer stemness and risk of metastatic progression in lung adenocarcinoma. Moreover, we highlight P4HA2 as a potential target for uncoupling ECM signals that support cancer stemness.

**Keywords** cancer stem cells; extracellular matrix; lung cancer; RASSF1A; stiffness

**Subject Categories** Cancer; Cell Adhesion, Polarity & Cytoskeleton; Stem Cells

The EMBO Journal (2019) 38: e100532

## Introduction

Cellular heterogeneity within the tumour microenvironment has been reported as a general property of solid cancers (Hanahan & Weinberg, 2011; Marusyk *et al*, 2014). The population of cells referred to as cancer stem cells (CSCs) exhibit extensive self-renewal abilities, multi-potent differentiation and increased metastatic tumour formation (Al-Hajj *et al*, 2003; Ponti *et al*, 2005; Sales *et al*, 2007; Wang *et al*, 2013). Although CSCs represent < 1% of total population of a tumour, they represent a major contributing factor to radio or conventional chemotherapy resistance and aggressive progression (Salcido *et al*, 2010; Kaseb *et al*, 2016). Dedifferentiation of cancer cells to a more pluripotent-like state results in the appearance of CSCs during malignant tumorigenesis (Codony-Servat *et al*, 2016); however, the mechanism behind this process in solid tumours remains to be elucidated. Evidence suggests that interaction with the surrounding microenvironmental niche contributes to the conversion of non-stem cancer cells to CSCs (Chaffer *et al*, 2011; Gupta *et al*, 2011). Both biochemical and physical signals from the tumour microenvironment can modulate properties of CSCs endowing them with the potential to adapt to the emerging cancer niche (Chen *et al*, 2012; Driessens *et al*, 2012; Schepers *et al*, 2012; Schwitalla *et al*, 2013). Mechanical properties provided by extracellular matrix (ECM), e.g. increased tissue stiffness, enhanced deposition or crosslinking, have been described to influence CSC plasticity and trigger stemness in non-stem cancer cells (Wong & Rustgi, 2013; Ye *et al*, 2014). In line with promotion of CSCs, increased matrix stiffness within tumour tissue also correlates with elevated cancer invasion, migration and metastatic spreading (Morrison & Spradling, 2008; Lane *et al*, 2014; Scadden, 2014; Turner & Dalby, 2014). Mechanotransduction from the ECM serves as upstream regulator of the Hippo pathway transcription factors YAP/TAZ (Piccolo *et al*, 2014), recently reported as an essential component for formation of lung CSCs (Halder & Johnson, 2011; Tremblay & Camargo, 2012; Park *et al*, 2018). Nuclear YAP1 is responsible for numerous oncogenic properties of tumour cells and is restricted by the Hippo pathway-mediated phosphorylation on serine127 (YAP1-pS127) (Zhao *et al*, 2008; Zanconato *et al*, 2015). RASSF1A is a key regulator of Hippo signalling in humans, and loss of expression has been correlated with reduced YAP1-pS127 across multiple clinical cohorts, including lung (Vlahov *et al*, 2015). Independently, RASSF1A has been extensively validated as a tumour suppressor in lung cancer where promoter methylation-associated gene silencing correlates with poor progression and overall survival (Burbee *et al*, 2001; Lee *et al*, 2001; Neyaz *et al*, 2008; Pallarés *et al*, 2008; Grawenda &

1 Department of Oncology, University of Oxford, Oxford, UK
2 Oxford Institute for Radiation Oncology, University of Oxford, Oxford, UK
3 TDI Mass Spectrometry Laboratory, Nuffield Department of Medicine, Target Discovery Institute University of Oxford, Oxford, UK
4 School of Chemistry, Physics and Mechanical Engineering, Queensland University of Technology, Brisbane, Qld, Australia
5 Systems Biology Ireland, University College Dublin, Dublin 4, Ireland
  *Corresponding author. Tel: +44 1865617321; E-mail: eric.oneill@oncology.ox.ac.uk

O'Neill, 2015). Although RASSF1A has been studied *in vitro*, the precise consequence of loss of RASSF1A expression *in vivo* has been difficult to discern. Gene expression analysis of lung and breast cancers has recently provided insight as, in addition to YAP1 activation, embryonic stem cell (ESC) signatures are significantly elevated in human tumours lacking RASSF1A (Pefani *et al*, 2016).

To address the precise consequence of RASSF1A loss, we directly assessed tumour characteristics in an orthotopic lung tumour model and found that RASSF1A impedes tumour growth and metastatic dissemination. We demonstrated that loss of RASSF1A correlates with YAP1 driven expression of prolyl 4-hydroxylase alpha-2 (*P4HA2*) which supports collagen I deposition. Concomitantly, we found that high collagen deposition with associated elevation in tissue stiffness negatively correlates with RASSF1A expression *in vitro* and *in vivo*. Moreover, our data indicate that stiff ECM induces the pluripotency cassette (NANOG, OCT4 and SOX2) via β-catenin-YAP-associated transcription, which results in a stem-like cell population in RASSF1A null lung cancer cells. Interestingly, we also show that RASSF1A itself may potentially be mechanoresponsive as only suppresses YAP on soft ECM, but fails to do so on stiff ECM. Together, our study provides evidence that widespread clinical prognostic value attributed to RASSF1A epigenetic silencing is due to ECM remodelling associated with occurrence of CSCs. These findings offer clearer prognostic information for *RASSF1* methylation and new therapeutic opportunities to combat the underlying heterogeneity behind treatment failures.

# Results

## RASSF1A suppresses metastatic dissemination in lung adenocarcinoma

DNA methylation of the CpG island spanning the RASSF1A promoter has been widely appreciated to associate with poor clinical outcome of non-small cell lung cancer (Kim *et al*, 2003; Fischer *et al*, 2007). Surprisingly, evidence for similar prognostic association with RASSF1A mRNA has been lacking. To address this, we explored mRNA of lung cancer patients from The Cancer Genome Atlas (TCGA) where we found that RASSF1A expression positively correlates with a good prognosis and overall survival in lung adenocarcinoma patients but not in squamous cell carcinomas (Fig 1A). To address the physiological role of RASSF1A and its involvement during cancer progression, we constructed isogenic H1299 human lung adenocarcinoma cell lines (where *RASSF1A* is highly methylated) and transfected either with pcDNA3, referred as H1299[control], or stably expressing RASSF1A, referred as H1299[RASSF1A] (Fig 1B). As RASSF1A is one of the central scaffolds of Hippo pathway in mammalian cells (Matallanas *et al*, 2007), we found higher levels of LATS1-pS909 and YAP1-pS127 in H1299[RASSF1A] cells, signifying that Hippo pathway kinase is activated in the presence of RASSF1A (Fig 1B). Proliferation assays show that H1299[RASSF1A] maintains an intrinsic proliferation rates similar to the H1299[control] (Fig 1B, graph) and therefore can serve as an appropriate isogenic system to address RASSF1A-specific effects *in vivo*. To determine the physiological role of RASSF1A, we employed an orthotopic model to accurately recapitulate the natural tumour microenvironment, asses the relationship between mechanical and biological properties and allow monitoring

of metastatic dissemination from the lung primary tumour site (Fig 1C) (Boehle *et al*, 2000). Orthotopic mouse lung injection was performed by intrathoracic cell injection to avoid pneumothorax and mechanical damage as has been previously validated (Onn *et al*, 2003; Servais *et al*, 2012). H1299[control] and H1299[RASSF1A] isogenic stably transfected cells were injected into the left lung of mice and examined for tumour formation and metastatic events after 17 or 30 days (Figs 1D and E, and EV1A and B, Table EV1). Metastatic events were counted and quantified as lung surface nodules after mice were euthanized, as previously validated (Chen *et al*, 2015; Tan *et al*, 2016). At day 30, a total of 13/14 mice injected from two different experiments with separate pools of H1299[control] cells (in which RASSF1A is not expressed) developed clear evidence of primary tumours in left lungs (Figs 1D and EV1A). Moreover, 13/14 H1299[control] mice also developed ipsilateral (left lung) metastases and 12/14 contralateral metastases in the right lung (Fig 1E, Table EV1). Surprisingly, 8/12 mice injected with H1299[RASSF1A] cells developed primary tumours, but there was striking suppression of metastatic events in the both the ipsilateral (left) and contralateral (right) lungs at day 30 (Figs 1D and E, and EV1A, Table EV1). These data suggest that RASSF1A may only have a minor suppressive effect on primary tumour growth, but loss of expression has a significant effect on metastatic events. When mice were sacrificed earlier (day 17), there was no difference in primary tumours and no metastases were apparent at this time, suggesting that metastases are unlikely to be an early event but a consequence of dissemination from a mature primary tumour (Fig EV1B). Primary tumours were also accompanied by production of a liquid oedema (Movies EV1 and EV2) around lungs indicating possible inflammation (Matthay, 2014); however, staining with the macrophage marker F4/80 indicated no significant difference in an inflammatory response between H1299[control] and H1299[RASSF1A] (Fig EV1C). We next selected HOP92 cells that express endogenous RASSF1A (Pefani *et al*, 2016) and constructed a comparable isogenic derivative expressing shRASSF1A to silence expression. Inactivation of Hippo pathway was confirmed in HOP92[shRASSF1A] cells (Fig EV1D), and, in contrast to H1299 cells, this was associated with an elevated proliferation rate *in vitro* (Fig EV1D). HOP92shcontrol cells were injected into the left lung of mice but resulted in limited formation of primary tumours at day 30 (1/7 mice, 16%), which was increased upon silencing of RASSF1A (3/7 mice, 42%) with evidence of at least one metastatic event (Fig EV1E, Table EV2). Taken together, these data imply that the adverse prognosis associated with reduced RASSF1A expression is most likely to be due to increased metastatic dissemination.

## RASSF1A suppresses P4HA2 expression and collagen I deposition

Invasion from primary tumour is not an autonomous process of cancer cells, but involves cross-talk with surrounding stromal tissue and ECM (Xu *et al*, 2009). The absence of RASSF1A in primary tumours appears to support metastatic development; therefore, we hypothesized that the H1299[control] microenvironment may be distinct. To identify candidates involved, we performed mass spectrometry on isolated ECM from both H1299[control] and H1299[RASSF1A] cells. As expected, a variety of ECM components were produced by both cells, e.g. collagens, laminins and fibronectin; however, ECM isolated from H1299[control] exclusively contained trophoblast glycoprotein (TPBG), laminin-beta2 (LAMB2), prolyl 4-hydroxylase

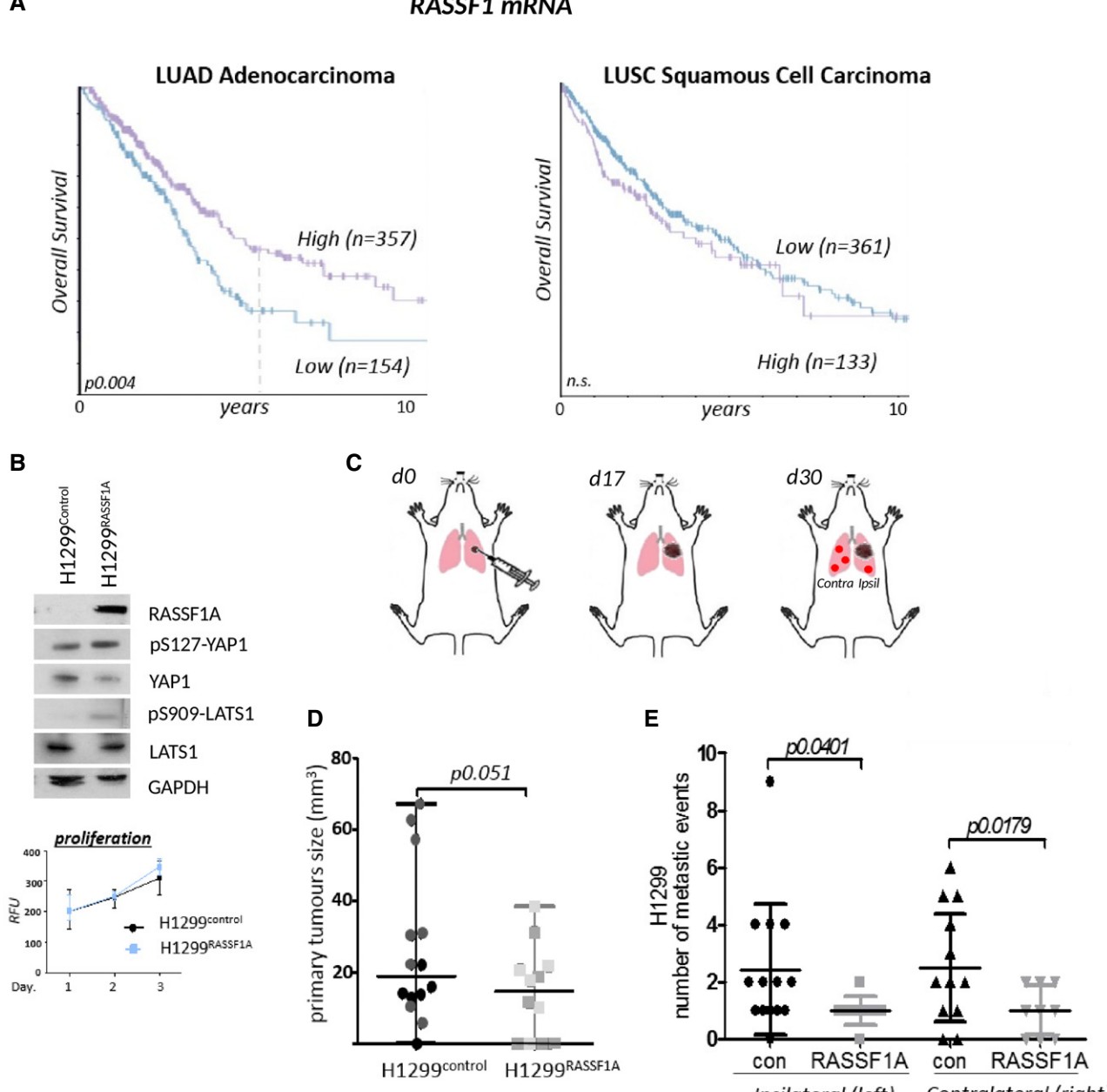

**A** *RASSF1 mRNA*

**Figure 1. RASSF1A suppresses metastasis in lung adenocarcinoma.**

A   Kaplan–Meier curves for overall survival (OS) in lung adenocarcinoma TCGA_LUAD (RASSF1 mRNA high/low cutoff FKPM 5.85) and squamous cell carcinoma patients TCGA_LUSC (RASSF1 mRNA high/low cutoff FKPM 6.52). Significance derived from log-rank test.

B   Western blot with indicated antibodies of isogenic H1299 cells stably transfected with either empty vector pcDNA3 (H1299$^{control}$) or RASSF1A (H1299$^{RASSF1A}$). Bottom: cell proliferation resazurin assay. ($n$ = 2). Error bars represent mean ± SEM.

C   Cartoon of lung adenocarcinoma orthotopic injection, with sites of primary and metastatic tumours indicated (ipsil = ipsilateral, same lung as primary tumour; contra = contralateral in the opposite lung). Mice were euthanized to collect the lungs at day 17 or day 30 after inoculation with tumours cells into the left lung.

D   Size of lung primary tumours measured at day 30 after lung orthotopic injection either with H1299$^{control}$ ($n$ = 14 mice per group; experimental groups of 6 and 8 different shading within the groups represents two independent experiments) or with H1299$^{RASSF1A}$ cells stably re-expressing RASSF1A ($n$ = 12 mice per group; experimental groups of 6 and 6). The size of primary tumours was measured by MRI software ITK-SNAP. Statistical significance via 2-tailed Student's *t*-test. Error bars represent mean ± SEM.

E   Visual assessment quantification of metastatic events on the ipsilateral (left) and or contralateral (right) lungs, generated by H1299$^{control}$ or H1299$^{RASSF1A}$ at day 30. Result was calculated from two independent *in vivo* experiments (as in D). Graph shows significant decreasing of metastases when lungs were injected with H1299$^{RASSF1A}$. Statistical significance via 2-tailed Student's *t*-test. Error bars represent mean ± SEM.

Source data are available online for this figure.

alpha-2 (P4HA2), ADAMTSL5 and SERPINB9 (Fig 2A). Interestingly, high mRNA expression levels of ECM components identified from H1299[control] also significantly correlate with poor prognosis in lung adenocarcinoma patients in line with low RASSF1A expression (Fig 2B). Reciprocally, we isolated MRX7A, SDF2, TFPI2 and TIMP1 specifically from H1299[RASSF1A] ECM which, conversely to above, are positively correlated with overall survival in lung adenocarcinoma (Fig 2B) or breast cancer patients (Fig EV2A). P4HA2 is the key enzyme involved in the collagen-specific posttranslational modification, catalysing the formation of 4-hydroxyproline residues, crucial for proper collagen folding and fibre stabilization (Myllyharju, 2003, 2008). We further evaluated TCGA data and find that high mRNA expression levels of P4HA2 significantly correlate with worse clinicopathological prognosis and survival outcome across many solid cancers (Fig EV2B). Immunohistochemical (IHC) staining of P4HA2 levels shows its increased expression in mouse primary tumours (peri-nuclear localization due to processing of collagen in the ER; Human Protein Atlas), generated by H1299[control] cells (Fig 2C). To ascertain if P4HA2 levels may have a functional consequence, we first measured the total enrichment of total hydroxyproline in cells and found elevated levels in H1299[control] cells (Fig 2D). P4HA2 activity leads to increased collagen fibres, and concomitantly, increased collagen I levels were observed in H1299[control] cells compared with H1299[RASSF1A](Fig 2E), suggesting these cells produced highly stable collagen fibres, as known for the hydroxyprolinated collagen (Mizuno et al, 2004). To evaluate whether this was due to P4HA2 activity, collagen deposition was measured by Western blot and by immunofluorescence (IF) in the presence of siRNA targeting P4HA2 mRNA and the prolyl 4-hydroxylase inhibitor 1,4-DihydroPhenonthrolin-Carboxylic Acid (DPCA) to restrict activity (Fig 2E–G). As expected, collagen deposition in H1299[control] cells is dependent on P4HA2; moreover, the lack of additive effect with siP4HA2 plus DPCA suggests that the inhibitor is unlikely to be working through an independent mechanism (Fig 2G). H1299[RASSF1A] cells not only display reduced collagen, but also reduced levels of P4HA2 mRNA, indicating transcriptional suppression as a possible reason for loss of expression (Fig EV2C and D). To determine if P4HA2 levels and effects on collagen are stimulated by RASSF1A loss, we measured P4HA2 and collagen I levels in HOP92 by IF and found levels to be induced and clear evidence for stable collagen in the absence of RASSF1A (Fig. 2H and I). As an additional control, we suppressed RASSF1A with siRNA in HeLa cells, which express high levels of RASSF1A, and observed a similar induction in P4HA2 and collagen by IF (Fig 2J).

### RASSF1A alters invasion and properties of ECM

To address whether our *in vivo* data were related to alterations in collagen deposition *in vitro,* we next investigated whether invasive potential of H1299[RASSF1A] was altered compared with H1299[control] (Fig 3A). RASSF1A-expressing cells demonstrated a decreased ability to invade through three-dimensional (3D) collagen compared with H1299[control] (Fig 3A). However, since complex collagen I matrix only mimics parenchymal tissue (Liotta, 1986), we additionally used a Matrigel matrix, highly enriched with laminins, to investigate the effect of P4HA2 depletion on invasion through basement membrane. We found that invasion of H1299[control] cells through Matrigel is also dependent on P4HA2, as knockdown or inhibition significantly

reduced invasion to an equivalent level of H1299[RASSF1A] (Fig 3B). To support the hypothesis, we tested HOP92 cells and found that suppression of RASSF1A mRNA increased invasion (Fig 3C). Tissue remodelling and ECM alignment are major processes that facilitate cancer cell invasion into surrounded tissue (Miron-Mendoza et al, 2008; Gehler et al, 2013; Han et al, 2016). To address this as a functional consequence of collagen fibre stability, we performed a collagen gel contraction assay that assesses the ability to reorganize a 3D-collagen matrix *in vitro*. After 4 days, extensive matrix remodelling and contraction of collagen plugs by H1299[control] cells were apparent, whereas collagen plugs containing H1299[RASSF1A] cells remained unaltered (Fig 3D). Moreover, remodelling of collagen plugs was impaired by blocking P4HA2 activity in H1299[control] cells (Fig 3D), suggesting that P4HA2 is essential for the ability of H1299[control] to reorganize surrounding tissue during invasion. To further evaluate how H1299 cancer cells respond to the mechanical properties of three-dimensional extracellular matrix, we generated spheroids and embedded these into a collagen matrix (Fig 3E). YAP is a mechanical sensor of ECM stiffness, also regulated by the Hippo pathway with a potential to activate P4HA2 transcription similar to that of Taz (Matallanas et al, 2007; Dupont et al, 2011; Piersma et al, 2015). We observed strong YAP nuclear localization in H1299[control] 3D spheroids, whereas, in line (Ueno et al, 2014; Papageorgis et al, 2015)with its role in Hippo pathway activation, expression of RASSF1A reduced YAP nuclear accumulation (Fig 3E). Moreover, the P4HA2 expression we observe in H1299[control] cells is YAP dependent (Fig 3F). These results led us to question whether the constitutive activation of YAP was a consequence of RASSF1A loss or a result of increased collagen I deposition. To address this, we constructed adoptive ECM experiments where decellularized ECM from collagen containing H1299[control] cells (control ECM) was used to determine whether ECM alone is sufficient to promote nuclear YAP in re-seeded H1299[RASSF1A] cells. Re-seeding of H1299[control] or H1299[RASSF1A] onto their own ECM did not affect YAP localization, but interestingly the predominantly cytoplasmic staining of YAP in H1299[RASSF1A] became concentrated in the nucleus when plated on control ECM (Fig 3G). In order to test the effect of reducing collagen deposition and ECM in the absence of RASSF1A, we used decellularized matrix from H1299[control] cells which had been treated with siP4HA2 or DCPA (as in Fig 2G). Cells grown in this matrix retained YAP1 cytoplasmic localization, suggesting that ECM is dominant over Hippo pathway signalling but also that RASSF1A itself may respond to ECM to suppress YAP (Fig 3G).

### Loss of RASSF1A is associated with more organized ECM and stiffness *in vitro* and *in vivo*

Previously reported studies demonstrated that cancer cell invasion is promoted by increased rigidity of the tumour microenvironment and tissue tension (Provenzano et al, 2006; Leventals et al, 2009). We next employed atomic force microscopy (AFM) analysis to measure tension and found that H1299[control] cells produce stiffer ECM accompanied by production of highly dense fibrillary network *in vitro* (Fig 4A and B). Consistent with these data, *in vivo* topographic analyses of primary lung tumours generated by H1299[control] cells displayed elevated stromal stiffness (16 kPa) that positively correlated with a more highly compact extracellular network compared with H1299[RASSF1A] (Fig 4C–E). Collagen is the main component of ECM responsible for network formation within the

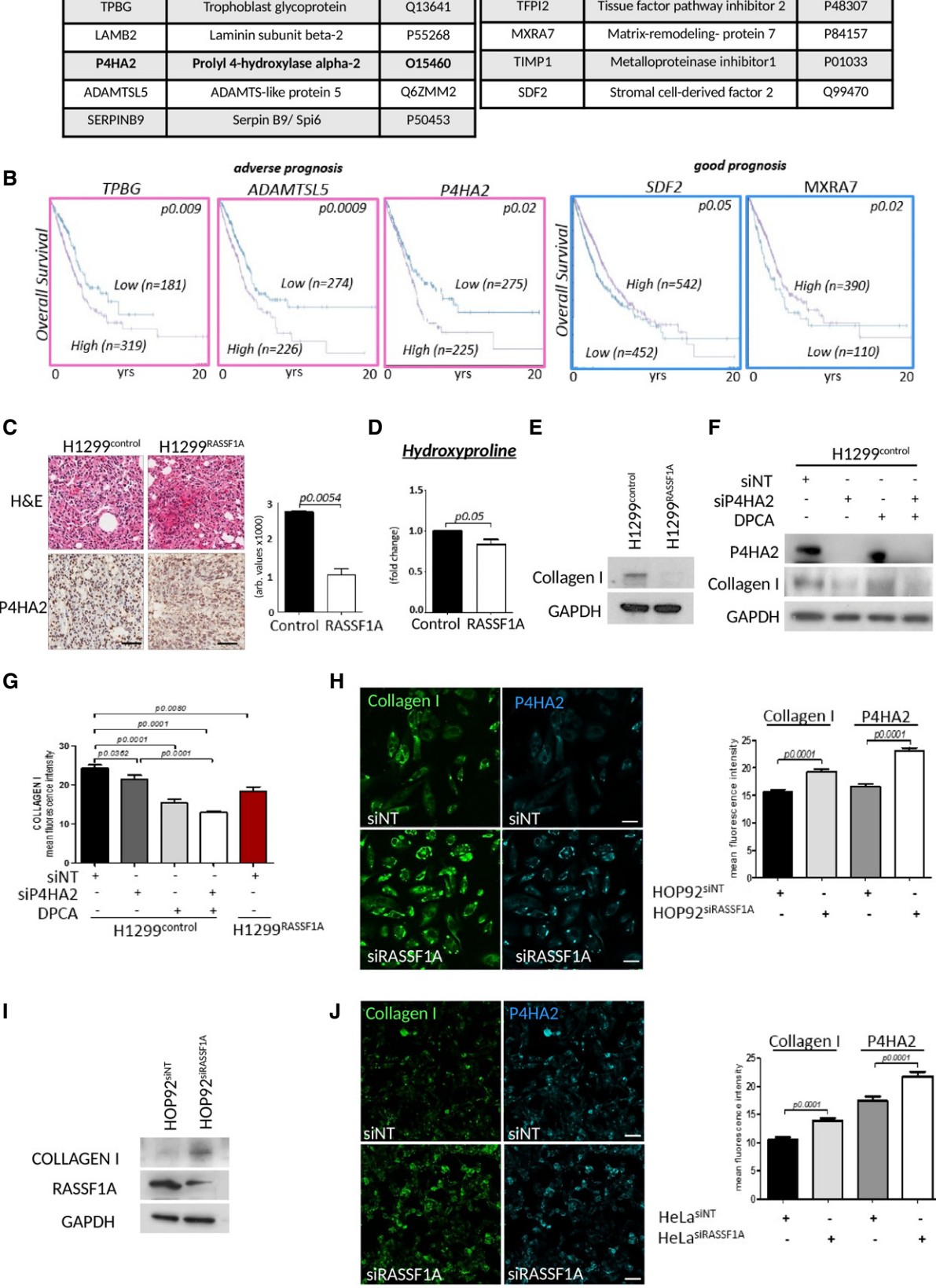

Figure 2.

◀

**Figure 2.  RASSF1A methylation is associated with increased P4HA2 and collagen I deposition.**

A   Mass spectrometry analysis of proteins purified from extracellular matrix isolated from H1299[control] and H1299[RASSF1A] lung adenocarcinoma cell lines with summary of results. Only proteins identified with two or more peptides were taken into consideration. The resulting list of proteins was restricted to ECM proteins (*n* = 3).

B   Kaplan–Meier plots depicting prognosis in TCGA_LUAD (lung adenocarcinoma) with high and low mRNA expression of ECM proteins revealed by proteomics. Significance derived from log-rank test.

C   Representative images of H&E and IHC staining for P4HA2, and quantification (bars) of two independent regions of *n* = 5 H1299[control] and *n* = 3 H1299[RASSF1A] primary lung tumours. Scale bars: 100 μm.

D   Hydroxyproline assay (Cell biolabs Inc.) for total hydroxyproline activity of cells.

E   Western blot indicating levels of collagen I in H1299[control] and H1299[RASSF1A] cells.

F   Western blot indicating levels of collagen I in H1299[control] cells in the presence of (4 μM) prolyl 4-hydroxylase inhibitor (DPCA), siRNA against P4HA2 or combination.

G   Quantification of immunofluorescence staining of collagen I in H1299[control] cells in the presence of (4 μM) prolyl 4-hydroxylase inhibitor (DPCA), siRNA against P4HA2 or combination. H1299[RASSF1A] cell line was used as negative control for comparison. Representative images in Fig EV2C.

H   Immunofluorescence staining for collagen I and P4HA2 and quantification (right graph) in HOP92 cells after siRNA against RASSF1A. Scale bars: 20 μm.

I   Western blot analyses in HOP92 cells showing increased expression of collagen I after siRNA against RASSF1A.

J   Immunofluorescence staining for collagen I and P4HA2 and quantification (right graph) in HeLa cells after siRNA against RASSF1A. Scale bars: 20 μm.

Data information: Unless otherwise indicated, all statistical analyses were performed using Student's *t*-test (two-tailed) of *n* = 3 experiments, and error bars represent the mean ± SEM.
Source data are available online for this figure.

tumour microenvironment (Provenzano *et al*, 2006). P4HA2 is known to have a major effect on physical properties of tumour-associated ECM, which in turn leads to increased stiffness during cancer progression (Provenzano *et al*, 2006; Levental *et al*, 2009). To investigate whether loss of RASSF1A in H1299[control] cells is associated with collagen organization, we embedded spheroids into a non-crosslinked collagen matrix (2 mg/ml) and examined collagen deposition by second harmonic generation microscopy (SHG), which only detects native, self-assembled polarized collagen fibres with non-centrosymmetric molecular structure (Chen *et al*, 2012). We found that cells from H1299[control] spheroids locally produce long, very organized collagen fibres that also appear to serve as tracks for invading cells from the central spheroid (Fig 4F red arrows, G), as noted previously (Han *et al*, 2016). As expected, long collagen fibres were not present upon treatment with siP4HA2 or DPCA and a disorganized mesh identical to the collagen network produced by H1299[RASSF1A] spheroids was apparent (Fig 4F and G). Correspondingly, SHG microscopy of tumours generated by H1299[control] showed significantly greater level of organized collagen fibres compared with H1299[RASSF1A], which similar to *in vitro* observations, remained a disperse organization with no unifying pattern (Fig 4H). Intriguingly, pre-metastatic stage day 17 lungs showed that ipsilateral (left) lungs injected with H1299[control] cells displayed widespread organized collagen deposition and organization in the ipsilateral lung away from the site of injection in contrast to the contralateral lung (Fig EV3A), which is similar to pre-metastatic niche deposition (Fig EV3A) (Fang *et al*, 2014). Desmoplasia, the intense fibrotic response characterized by the formation of very compact collagen-enriched ECM, greatly contributes to aggressiveness during cancer progression (Liu *et al*, 2010) and has been correlated to worse clinical outcome in various cancers (Ueno *et al*, 2014; Papageorgis *et al*, 2015). Examination of primary tumours for fibrotic tissue with *picrosirius red* staining showed that H1299[control] lung tumours displayed an extended fibrotic area not observed in H1299[RASSF1A] (Fig 4I). Taken together, our data indicate that YAP1 drives P4HA2 expression in RASSF1A-methylated tumours, resulting in increased organization of collagen and elevated stiffness of the tumour microenvironment. We also demonstrate that loss of RASSF1A and increased P4HA2 activity are associated with dissemination of cancer cells, which is in line with previous observations relating stiffness and metastasis.

## ECM stiffness is important for NANOG expression and nuclear translocation

The growing evidence that deregulation of ECM dynamics plays an essential role in the generation of a tumour stem cell niche and generation of cancer stem cells (Bissell & Labarge, 2005; Plaks *et al*, 2015) led us to question, whether the mechanical properties of ECM can induce stemness. Recently, the stem cell transcription factor NANOG has been described in clinical studies to be associated with appearance of cancer stem cells (Lin *et al*, 2005; Liu *et al*, 2017). We plated cells on custom made soft wells (Matrigen), coated with collagen matrix of defined stiffness and find that cells lacking RASSF1A constitutively induce re-programming to a cancer stem cell-like state (Fig 5A). Strikingly, we see that H1299[control] cells induce not only expression, but also nuclear translocation of NANOG when cultured on soft ECM (0.5 kPa) which H1299[RASSF1A] cells fail to do. However, elevating stiffness of ECM (4 kPa) to compensate for the lack of intrinsic stiff ECM with H1299[RASSF1A] readily increased NANOG levels (Fig 5A). Surprisingly, H1299[con] cells grown on extremely stiff ECM (25 kPa) retained cytoplasmic localization (Fig 5A) equivalent to cells grown on two-dimensional stiff glass surface without matrix (Fig 5B), while NANOG localization in H1299[RASSF1A] cells appeared less affected. To validate the relationship between cancer stemness and ECM stiffness, we investigated expression of the *bona fide* cancer stem cell marker CD133 (Alamgeer *et al*, 2013) and again showed positive correlation between increased ECM stiffness (4kPA) and greater expression in H1299[control] but not in H1299[RASSF1A] cells, suggesting that RASSF1A prevents cancer stemness in soft ECM (Fig 5C). As observed for NANOG, CD133 expression was not increased when H1299[control] and H1299[RASSF1A] cells were growing on very stiff (25 kPa) collagen matrix (Fig 5C). We hypothesize that extremely stiff ECM may lock the conformation of ECM molecules, preventing exposure of binding sites such as integrins and therefore reducing the ability to respond to ECM (Doyle & Yamada, 2016). To support the role of importance of collagen concentration in activation of stemness, we embedded H1299[control] and H1299[RASSF1A] single cells into a 3D matrix with increasing collagen concentrations (2, 2.5 and 3 mg/ml). H1299[control] cells exhibit high NANOG nuclear localization whereas H1299[RASSF1A] did not, as observed for soft ECM above, and again elevated collagen concentration overrides hippo

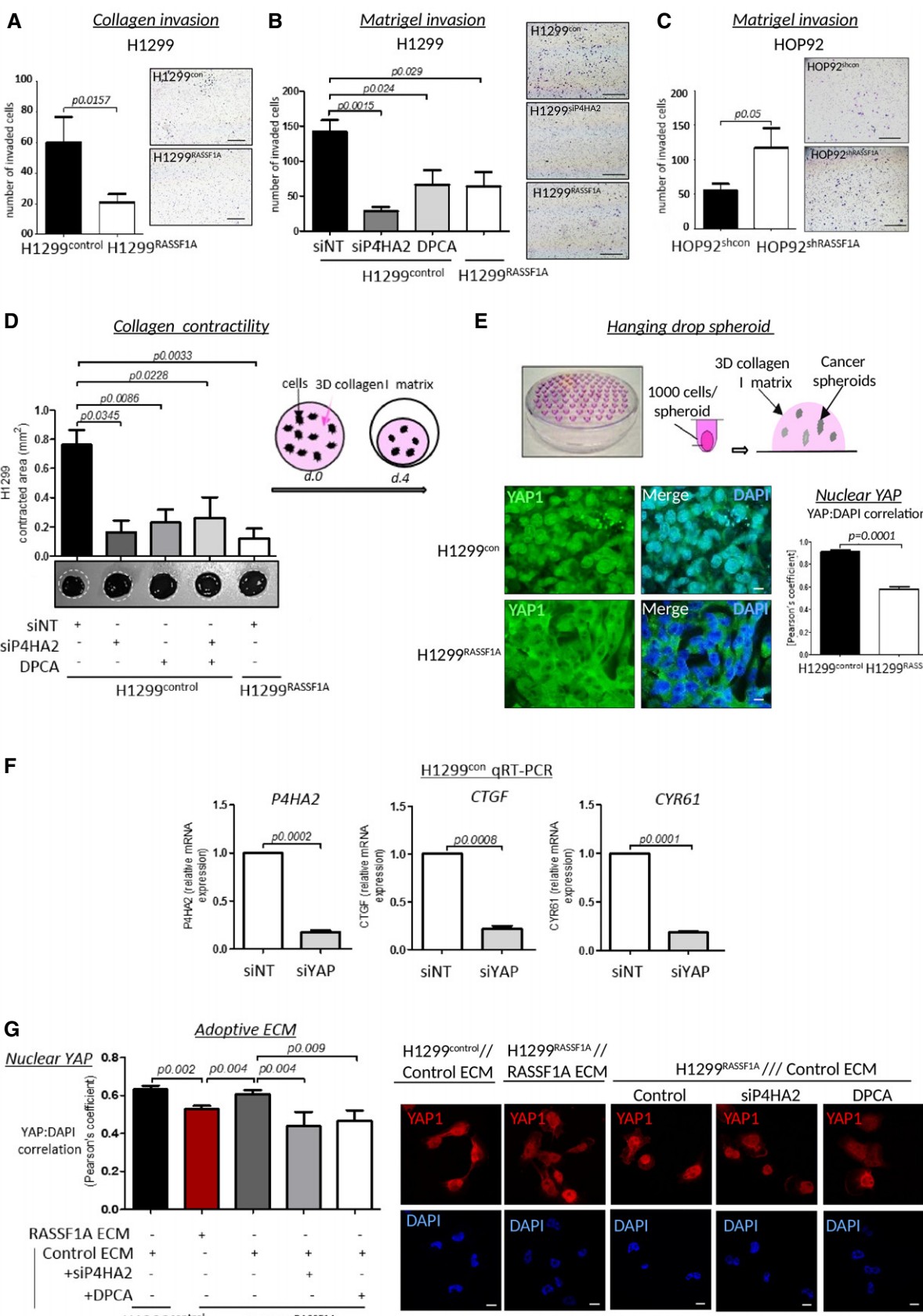

**Figure 3.**

**Figure 3.  Cells lacking RASSF1A alter ECM and are more invasive.**

A   Invasion of H1299[control] and H1299[RASSF1A] cells through three-dimensional collagen matrix coated inserts, over 24 h (*n* = 3). Scale bars: 100 μm.

B   Representative images and quantification of H1299 cells treated with siNT, siRNA against P4HA2 or in the presence of the prolyl 4-hydroxylase inhibitor DPCA (4 μM), and allowed to invade for 24 h through a three-dimensional Matrigel matrix Boyden chamber. Scale bars: 100 μm.

C   Representative images and quantification of HOP92 cells stably transfected with shcontrol or shRASSF1A and allowed to invade for 24 h through a three-dimensional Matrigel matrix Boyden chamber. Scale bars: 100 μm.

D   3D collagen contraction assay: $5 \times 10^5$/ml cells were embedded into collagen rat tail I matrix (2 mg/ml) and analysed after 4 days. White circles indicate diameter of gel plugs at time 0. Representative bright-field images (bottom) with quantification, showing the effect of H1299[RASSF1A] or H1299[control] cells on collagen gel remodelling and after cells pre-treatment with siRNA against P4HA2, 4 μM DPCA or combination as indicated.

E   Upper image: Cartoon of hanging drops method for spheroids formation and embedding collagen rat tail I matrix (2 mg/ml). Bottom: Representative immunofluorescence images of YAP distribution and its nuclear quantification (right bars) by Pearson's coefficient for correlation of YAP and DAPI co-staining in H1299[control] and H1299[RASSF1A] 3D spheroids grown in 3D collagen matrix (2 mg/ml). Scale bars: 10 μm.

F   Relative mRNA expression levels of P4HA2 and YAP target genes CYR61 and CTGF in H1299[control] cells in the presence of siNT or siRNA targeting YAP1.

G   Representative immunofluorescence images of YAP distribution (right) and its nuclear quantification by Pearson's coefficient for correlation (left) of YAP and DAPI co-staining in H1299[control] and H1299[RASSF1A] plated on extracted ECM from H1299[control] or H1299[RASSF1A] cells with or without treatment with siP4HA2 or 4 μM DPCA. (Correlation, Pearson's coefficient). (*n* = 3, 300 cells per experiment) Scale bars: 10 μm.

Data information: Unless otherwise indicated, all statistical analyses were performed using Student's *t*-test 2-(tailed) of *n* = 3 experiments, and error bars represent the mean ± SEM.

pathway regulation in H1299[RASSF1A] cells to allow nuclear localization of NANOG (as seen for YAP above) (Fig EV4A). These data suggested that increased concentration of collagen matrix elevated the avidity of ECM for cell-ECM binding in single cells. To mimic a 3D-tumour environment, we next employed our spheroid model and found that ECM rigidity and high cell density of H1299[control] spheroids activated NANOG in the central core (Fig EV4B), but this was completely absent from H1299[RASSF1A], implying that the artificially stiff ECM that can activate NANOG in H1299[RASSF1A] (Fig 5A) is not achieved under more physiological conditions (Fig 5D). To understand contributing factors that may be supporting NANOG activation and cancer stemness, we also checked β-catenin (Valkenburg *et al*, 2011) and HIF-1α as a known activator of P4HA2 (Gilkes *et al*, 2013). The dimension of our 3D spheroids is at the oxygen diffusion limit (300 nm diameter) which therefore should not be hypoxic (Gilkes *et al*, 2013), and accordingly, we do not see HIF-1α staining, whereas β-catenin is readily expressed in H1299[control] spheroids and susceptible to inhibition of P4HA2 with DPCA (Fig EV4C and D). IHC staining of H1299[control] primary tumours supported these results as NANOG and high levels of nuclear YAP1 are apparent in H1299[control] tumours, while the active Hippo pathway in H1299[RASSF1A] tumours retains the majority of YAP1 in the cytoplasm and no NANOG staining is discernable (Fig 5E).

**P4HA2 regulates expression of the pluripotency cassette and cancer stem-like cells**

YAP is a key activator of cancer stem cells (CSCs) in various tumours (Basu-Roy *et al*, 2015; Kim *et al*, 2015). To determine whether disruption of Hippo pathway and nuclear activation of YAP1 is associated with initiation of pluripotency, we measured mRNA levels of *NANOG*, *OCT4* and *SOX2* and found higher levels in H1299[control] cells (Fig 6A). To confirm NANOG dependency on YAP, we monitored expression by IF in 3D collagen matrix (as above) and demonstrate complete loss after YAP knockdown in H1299[con](Fig 6B). The loss of apparent stemness upon restriction of YAP in H1299[control], via IF, was further supported by reduction in mRNA and protein levels of the pluripotency cassette (NANOG, OCT4 and SOX2) (Fig 6C and D). Since we propose that YAP1 mediates these effects though P4HA2-mediated collagen deposition and stiffness (Piersma *et al*, 2015), we next examined whether stemness is dependent on P4HA2. In line with the results presented above, treatment of 3D spheroids with DPCA abolished NANOG nuclear staining and reduced YAP1 (Fig 6E). Lysates from single cells embedded in 3D collagen also indicated reduction in total NANOG and YAP1 upon treatment with DPCA or siP4HA2, but surprisingly we found levels of β-catenin were also sensitive (Figs 6F and EV4D).

**Figure 4.  RASSF1A expression is associated with disorganized ECM and abrogated stiffness *in vitro* and *in vivo*.**

A   Bar graph representing quantification of stiffness measured by atomic force microscopy (Young modulus) of ECM area generated by H1299[control] or H1299[RASSAF1A] cells *in vitro*.

B   Representative atomic force microscopy (AFM) images showing organization of extracellular matrix generated by H1299[control] and H1299[RASSF1A] cells in 3D collagen matrices (2 mg/ml).

C   Bar graph representing quantification of stiffness by atomic force microscopy (Young modulus) of *n* = 5 H1299[control] and *n* = 3 H1299[RASSF1A] primary lung tumours on day 30.

D   Representative topographic images of stiffness map of H1299[control] and H1299[RASSF1A] lung primary tumours provided by AFM.

E   AFM images of ECM fibre organization within primary lung tumours generated by H1299[control] and H1299[RASSF1A] cells on day 30.

F   Second harmonic generation (SHG) representative images and quantification (bars) showing deposition and organization of collagen fibres produced by H1299[control] and H1299[RASSF1A] spheroids embedded in collagen I matrix (2 mg/ml), treated with siNT, siRNA against P4HA2 or 4 μM DPCA to restrict P4HA2 activity. Red arrowheads show highly organized, long collagen fibres. Scale bars: 20 μm.

G   SHG images of organization and quantification (bars) of collagen fractions in primary lung tumours and metastases at day 30. Scale bars: 10 μm.

H   Representative H&E and quantification (bars) of picrosirius red staining of two independent regions of *n* = 5 H1299[control] and *n* = 3 H1299[RASSF1A] primary lung tumours. Scale bars: 100 μm; zoom: 20 μm.

Data information: All statistical analyses were performed using Student's *t*-test (two-tailed) of *n* = 3 for *in vitro* experiments, and error bars represent the mean ± SEM.

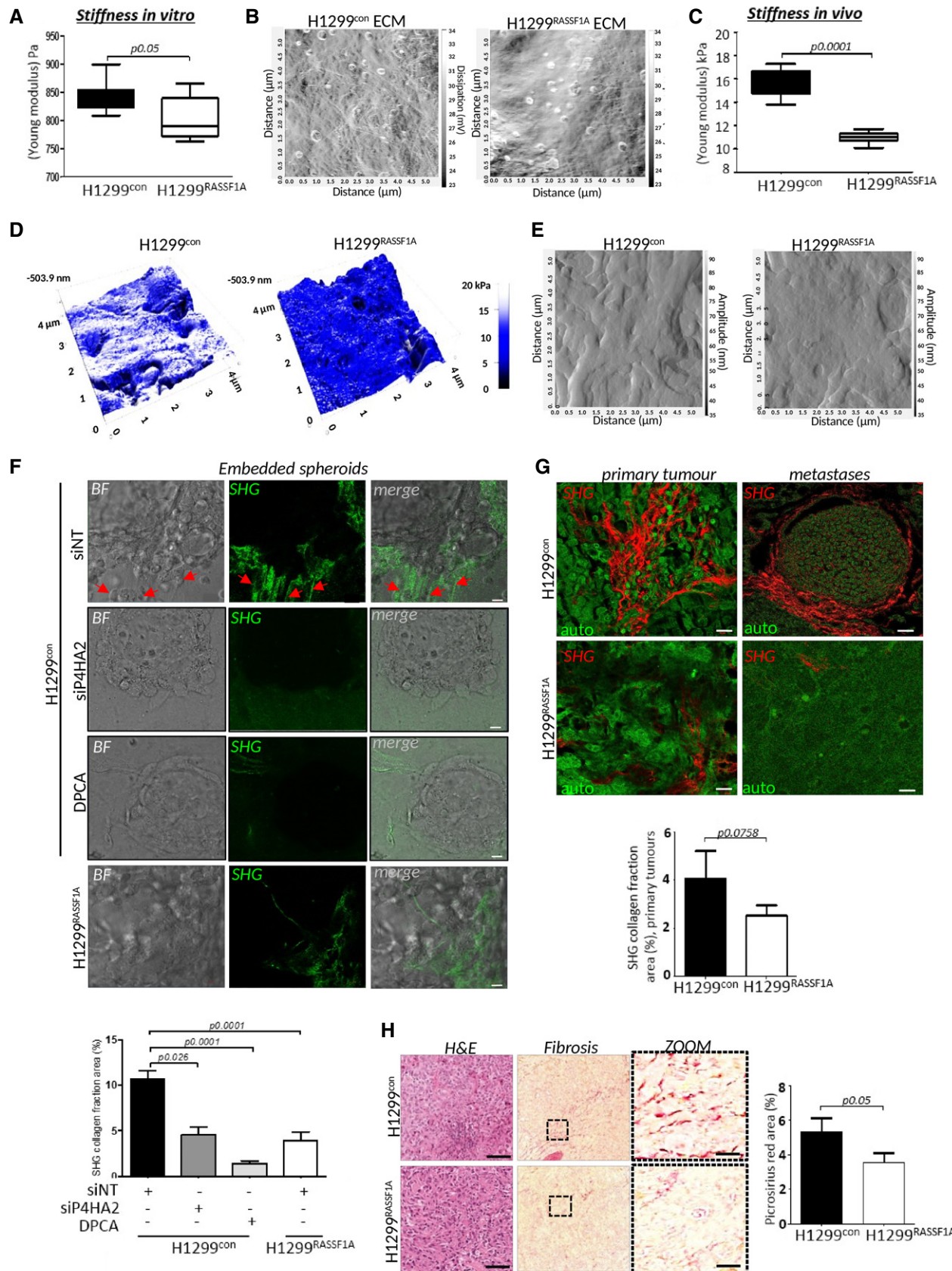

Figure 4.

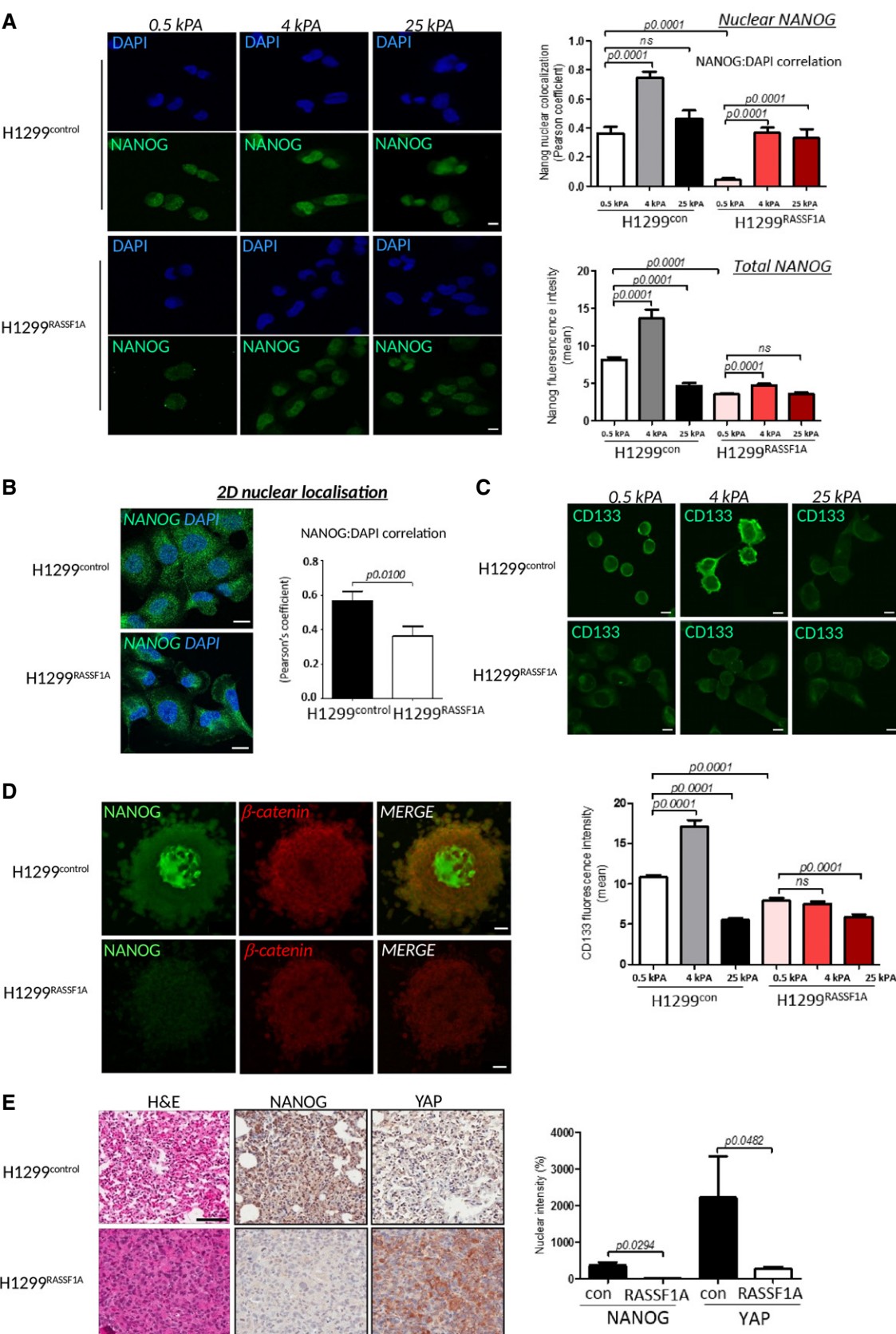

**Figure 5.**

**Figure 5. ECM stiffness is important for Nanog expression and its nuclear translocation.**

A  Left: Representative images of H1299 cells on 3D collagen wells with defined stiffness with NANOG (green) or DAPI (blue). Scale bars: 10 μm. Right: Quantification of NANOG:DAPI nuclear co-localization (upper graph) and total NANOG fluorescence intensity (bottom graph). Quantification of nuclear co-localization (n = 300 cells/experiment) is represented by Pearson's coefficient.

B  Representative immunofluorescence images show merge of NANOG (green), DAPI (blue) distribution in H1299[control] and H1299[RASSF1A] cells cultured on 2D glass. Scale bars: 10 μm. Right: Quantification of nuclear co-localization (n = 200 cells/experiment) is represented by Pearson's coefficient.

C  Representative immunofluorescence images of CD133 in H1299[control] and H1299[RASSF1A] cells grown on 3D collagen wells with defined stiffness. Scale bars: 10 μm. Bottom: Quantification of n = 200 cells/experiment.

D  Representative images of immunofluorescence staining for the pluripotency marker NANOG and β-catenin in three-dimensional spheroids embedded in collagen matrix (2 mg/ml). Scale bars: 50 μm. Quantification in Fig EV4B.

E  Representative images of H&E and immunohistochemical staining for NANOG and YAP1 in primary lung tumours (day 17). Scale bars: 100 μm. Right: Graph bars represent quantification of NANOG and YAP1 staining based on strong (3+, 2+) nuclear intensity (%) for at least two independent regions of n = 4 H1299[control] and n = 2 H1299[RASSF1A] primary tumours at day 17.

Data information: Statistical significance was determined by Student's t-test (2-tailed) of n = 3 experiments unless otherwise stated. Error bars represent the mean ± SEM.

## P4HA2-mediated collagen synthesis impairs cancer cell differentiation

Re-expression of RASSF1A in H1299 lung adenocarcinoma cells is associated with reduced invasion, metastatic progression and inability to generate cancer stem-like cells, which provides a rationale for widely reported poor clinical outcomes in *RASSF1*-methylated tumours (Grawenda & O'Neill, 2015). Both *RASSF1*-methylated and low *RASSF1A* mRNA tumours display signatures of embryonic stem cells (Pefani *et al*, 2016) and are poorly differentiated (Grawenda & O'Neill, 2015). Moreover, RASSF1A directly promotes a differentiation programme in stem cells (Papaspyropoulos *et al*, 2018). To see whether H1299[RASSF1A] tumours are well-differentiated, we measured levels of TTF-1 (thyroid transcription factor 1) and Mucin 5B, characterized markers for terminal lung differentiation (Li *et al*, 2012) for which expression in tumours is correlated with better prognosis and survival in lung cancer patients (Saad *et al*, 2004; Boggaram, 2009; Nass *et al*, 2018). H1299[RASSF1A] cells and lung tumours display significantly higher levels of Mucin 5B and TTF-1 compared with H1299[con] (Fig 7A and B). Consistent with our hypothesis, we also observed greater expression of TTF1 and Mucin 5B in H1299[RASSF1A] spheroids embedded in collagen (Fig 7C). To further explore differentiation *in vitro*, we cultured H1299 spheroids on basement membrane matrix (Matrigel) and found that while H1299[control] spheroids maintained rounded morphology, associated with cancer stem cell phenotype (Yu *et al*, 2017), RASSF1A-expressing H1299 spheroids collapsed and formed branched structures reminiscent of a differentiated epithelium (Fig 7D, Movie EV3). These data support a model whereby RASSF1A and the Hippo pathway maintain differentiation status in tumours and prevent formation of CSCs, explaining the association of RASSF1A loss with aggressive lung adenocarcinoma.

## Discussion

Lung cancer has a dismal prognosis with only 10% of patients surviving > 5 years after diagnosis. Despite improvement in chemotherapy and intended surgery techniques lung cancer is the leading cause of cancer-related mortality. Identification of reliable prognostic markers would increase greater confidence in determining potential metastatic recurrence in lung cancer patients and provide appropriate treatment to increase overall survival (Yang

*et al*, 2005). RASSF1A is a tumour suppressor in non-small cell lung cancer, and its epigenetic silencing correlates with advanced disease, metastatic potential and adverse prognosis (Dammann *et al*, 2000, 2001; Dreijerink *et al*, 2001; Morrissey *et al*, 2001; Donninger *et al*, 2007). However, in-depth research studies supporting these clinicopathological data have been lacking. Using the data from TCGA, we showed for the first time the direct epigenetic association between mRNA expression levels of RASSF1A and survival in lung adenocarcinoma patients. In line with these observations, results from our *in vivo* experiments demonstrate that RASSF1A restricts tumour formation and decreases metastatic spreading in the lungs. Metastatic progression is a complex process supported by interaction between cancer cells and the tumour microenvironment (Hanahan & Weinberg, 2011). Collagen is also the major component of extracellular matrix which is directly involved in biophysical features of tumour microenvironment (Provenzano *et al*, 2006). P4HA2 is one of the key enzymes involved in collagen maturation that catalyses formation and stabilization of collagen fibres (Myllyharju, 2003). Our data suggest that deregulation of P4HA2 levels in *RASSF1*-methylated tumours is a key factor behind the poor prognostic value of these cancers. Moreover, we show that P4HA2 mRNA is similarly associated with overall survival in multiple solid tumours.

Mechanical properties of the ECM within tumour tissue greatly contribute to metastatic dissemination and are directly involved in the efficacy of conventional therapies (Liu *et al*, 2012). Moreover, cancer stromal stiffness is a crucial regulator of epithelial–mesenchymal transition that supports cancer progression (Leight *et al*, 2012). Interestingly, dynamic mechanical forces from the ECM have been described to regulate cancer cell behaviour (Handorf *et al*, 2015). In line with this idea, ECM deposition and proteins from cancer cell-derived ECM are critical regulators of cell proliferation, invasion and metastatic progression (Iyengar *et al*, 2005; Aguilera *et al*, 2014). Serving as a platform for cross-talk between cells and stroma, the ECM is an essential provider of physical scaffolds to maintain tissue architecture and tissue-specific function (Xu *et al*, 2009). As a result of increased P4HA2 levels in tumour cells, we find increased collagen deposition and ECM stiffness, which supports the appearance of cancer stem-like cells. Tumour heterogeneity and growth dynamics during cancer progression are associated with the plasticity of cancer cells which can convert into cancer stem cells (CSCs) (Balic *et al*, 2006; Mani *et al*, 2008), the major contributors to drug resistance and cancer recurrence (Jordan *et al*,

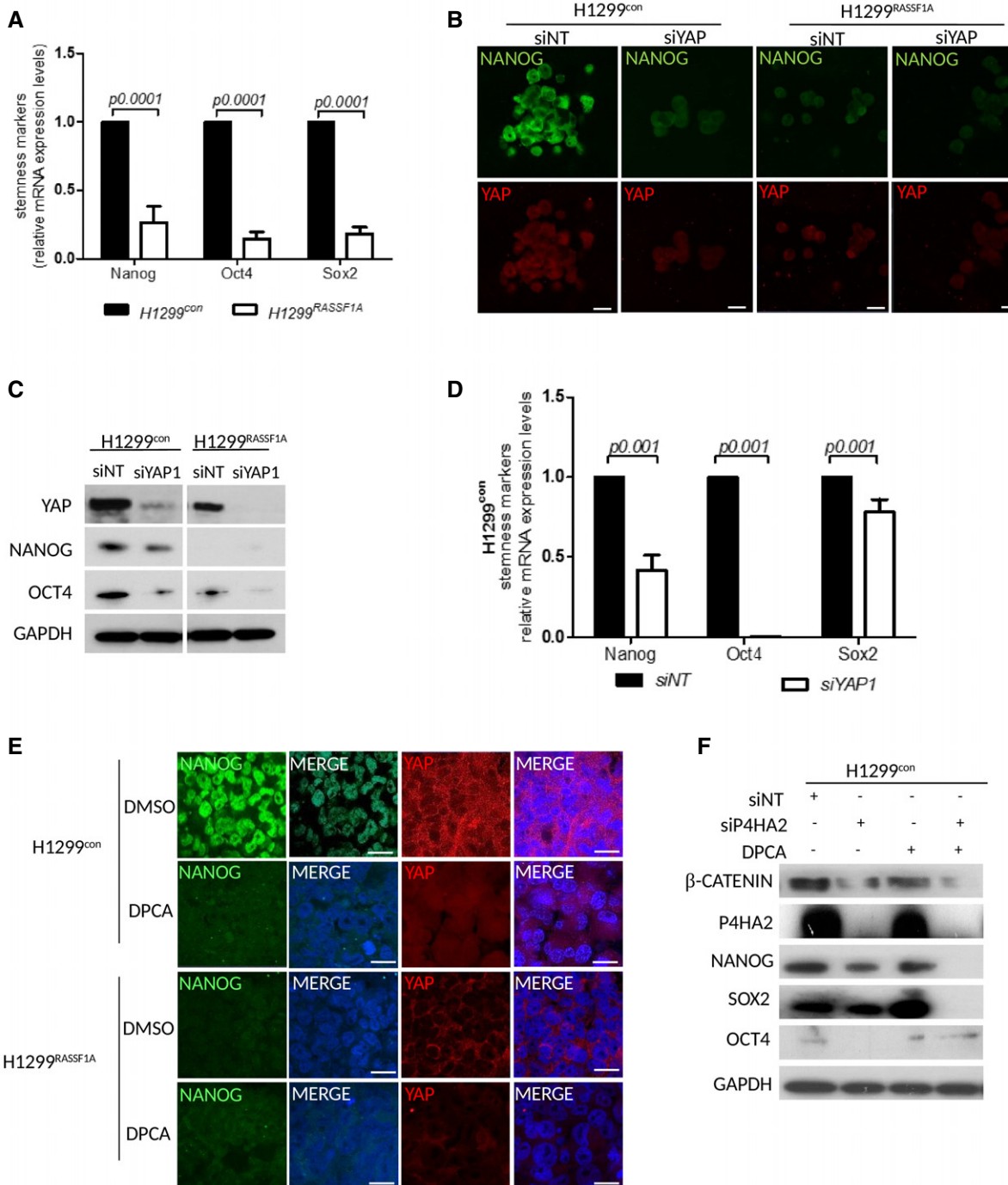

**Figure 6.  YAP regulates Nanog and OCT4 expression via P4HA2 in H1299[control] cells.**

A  Relative mRNA expression levels of NANOG, OCT4 and SOX2 in H1299[control] and H1299[RASSF1A] lung adenocarcinoma cells.

B  Representative immunofluorescence images of NANOG and YAP in the presence of siNT or siYAP in H1299[control] and H1299[RASSF1A] cells embedded and grown in three-dimensional collagen matrix (2 mg/ml). Scale bars: 10 μm.

C  Western blots of H1299[control] and H1299[RASSF1A] lysates for NANOG, SOX2 and OCT4 in the presence of siNT or siYAP.

D  Relative H1299[control] mRNA expression levels of NANOG, OCT4 and SOX2 in H1299[control] cells in the presence of siNT or siYAP.

E  Representative immunofluorescence images of H1299[control] and H1299[RASSF1A] spheroids embedded and grown in collagen matrix (2 mg/ml) and stained for NANOG (Green) and YAP (Red) with or without 4 μM DPCA. Scale bars: 10 μm.

F  Western blots of H1299[control] lysates for NANOG, SOX2 and OCT4 in the presence of siNT, siP4HA2 or 4 μM DPCA.

Data information: Statistical significance was determined by Student's *t*-test (2-tailed) of *n* = 3 experiments unless otherwise stated. Error bars represent the mean ± SEM.

Source data are available online for this figure.

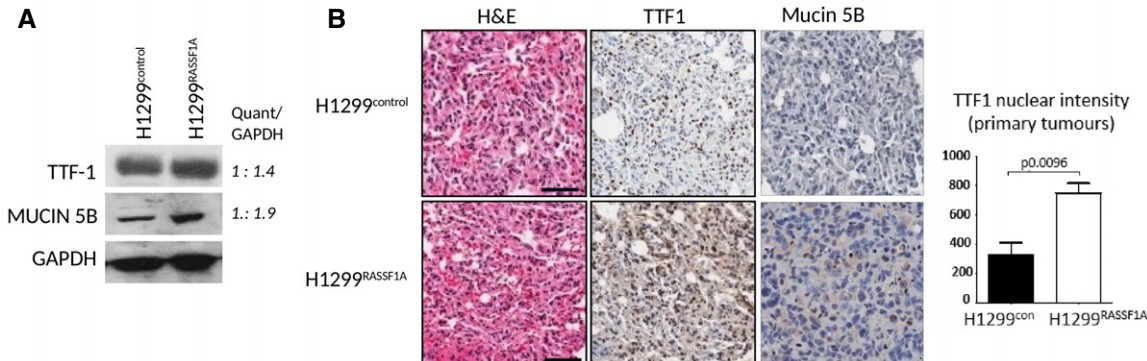

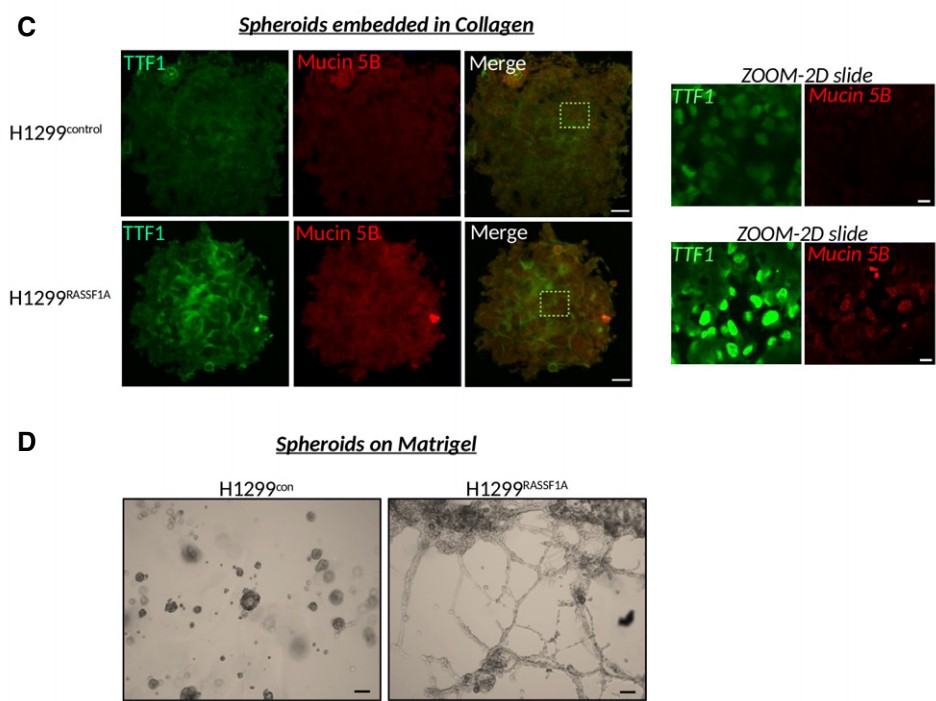

**Figure 7. P4HA2-mediated collagen synthesis attenuates cancer cell differentiation.**

A   Western blot analyses of expression levels of TTF-1 and Mucin 5B in H1299[control] and H1299[RASSF1A] cells with band intensity quantification ratio determine by ImageJ analyser.

B   Representative images of H&E and immunohistochemical staining for lung differentiation markers TTF-1 and Mucin 5B for at least two independent regions of n = 4 H1299[control] and n = 2 H1299[RASSF1A] primary tumours at day 17 with quantification (right) of TTF-1 based on nuclear intensity. Scale bars: 100 μm. Statistical significance was determined by Student's *t*-test (two-tailed). Error bars represent the mean ± SEM

C   Representative images of immunofluorescence staining of spheroids grown in collagen matrix (2 mg/ml) for differentiation markers TTF-1 (Green) and Mucin 5B (Red). Scale bars: 50 μm; Zoom: 10 μm.

D   Bright-field images of H1299[control] and H1299[RASSF1A] spheroid differentiation on Matrigel matrix after 24 h when 3D spheroids were seeded on three-dimensional substrate. Scale bars: 200 μm.

Source data are available online for this figure.

2006; Rahman *et al*, 2011). It was recently demonstrated that CSCs have a stem cell phenotype and express embryonic transcription factors OCT4, SOX2 and NANOG (Blassl *et al*, 2016). Moreover, deregulation of ECM dynamics has been reported to increase the population of cancer stem-like cells, although the mechanism has not been elucidated (Gattazzo *et al*, 2014). Here, we described that stiffness and ECM composition modulate NANOG expression and

are both important for maintenance of CSCs. Recent evidence indicates that YAP1 plays a crucial role in activation and maintenance of CSCs (Kim *et al*, 2015; Noto *et al*, 2017). However, the mechanism by which disruption of the Hippo pathway in cancers leads to induction of CSCs is not clear. Here, we report that reactivation of the Hippo pathway in *RASSF1*-methylated cells by re-expression of RASSF1A prevents YAP-mediated transcription of *P4HA2* and

ECM stiffness. The effect of ECM and YAP activity appears to combine to support transcription of the pluripotency cassette (*NANOG, OCT4 and SOX2*) which may also involve β-catenin to directly promote *OCT4* as we previously reported (Papaspyropoulos *et al*, 2018).

Our results also indicate that activated Hippo pathway signalling in cells grown on a soft collagen substrates (0.5 kPa or 1.5; 2 mg/ml) restricts phenotypic plasticity and appearance of CSCs. However, interestingly elevated ECM stiffness (4 kPA) or increased collagen concentrations (2.5 or 3 mg/ml) overrides this control. Intriguingly, this suggests that RASSF1A itself may respond to increased tension. Further investigation is needed to determine if this is the case and, if so, whether this occurs through interactions with Rho signalling or is more direct (Dubois *et al*, 2016; Lee *et al*, 2016). We also found that cells do not respond to very stiff collagen substrates (25 kPa), which we suggest may be due to loss of their ability to mechanically stretch or modify their fibres, required for appropriate cell signalling and mechanotransduction as previously observed (Kubow *et al*, 2015). This fits with reports that ECM can differentially affect integrin signalling depending on substrate type (Humphries *et al*, 2006), and mechanical forces generated by the ECM can activate or destroy binding sites on ECM-related proteins (Little *et al*, 2009). TTF-1 (also known as thyroid transcription factor 1) and Mucin 5B have been described as lung differentiation markers (Li *et al*, 2012) associated with better prognostic outcome in lung cancer patients (Saad *et al*, 2004; Boggaram, 2009). We have demonstrated that activation of the Hippo pathway in H1299^RASSF1A positively correlates with upregulation of TTF-1 and Mucin 5B *in vivo* and *in vitro*, suggesting that RASSF1A promotes differentiation of lung cancer cells, thus impairing tumour growth, metastatic progression and invasion *in vivo* and *in vitro*. Moreover, this also may indicate that differentiation status of the tumour may be affected by mechanical properties of the ECM.

Altogether, our results identify that high expression levels of RASSF1A can serve as a prognostic biomarker for lung adenocarcinoma patients. Loss or methylation of RASSF1A leads to constitutive YAP activity, which together with mechanical properties of extracellular matrix via P4HA2 drives cancer stem-like re-programming and metastatic progression in lung adenocarcinoma. Thus, this may explain how *RASSF1* methylation and tumour stiffness contribute to poor outcome in lung cancer patients, by reducing differentiation status and inducing pluripotency. Therefore, targeting P4HA2 may be an important new strategy of eradicating resistance of cancer stem cells during conventional tumour therapy.

# Materials and Methods

### Cell culture and drug treatments

Human lung adenocarcinoma cells H1299TetON-pcDNA3 or H1299TetON-RASSF1A under tetracycline promoter were previously described (Van Der Weyden *et al*, 2012; Yee *et al*, 2012). RASSF1A expression was induced by 1 μg/ml doxycycline for 24 h before experiments. H1299, HOP92 and HeLa cells were cultured in complete DMEM (Gipco) medium, supplemented with 10% foetal bovine serum (Sigma), penicillin/streptomycin (Sigma) and L-glutamine (Sigma) in 5% $CO_2$ humidified atmosphere at 37°C. Cells used

in experiments had been passaged fewer than 15 times. Activity of P4HA2 in cells was inhibited by using of 4 μM, 1.4-DPCA, a prolyl 4-hydroxylase inhibitor (Cayman Chemical Company) for 24 h.

### Generation of stable cell lines and siRNA interference

For generation of stable H1299 cell lines, we used human RASSF1A in pBABE system and pBABE-pcDNA3 control. H1299 cells were infected with lentiviruses carrying RASSF1A or control pcDNA3 packed in GP and VSVG plasmids for 48 h and then selected with puromycin (10 μg/ml) for 2 weeks. For RASSF1A silencing in HOP92 cells, we used human RASSF1 short hairpin RNA lentiviral particles (sc-44570V) and directly infected HOP92 regarding to manufacturer's instructions or with lentiviral particles carrying control pcDNA3 packed in GP and VSVG plasmids and incubated with target cells for 48 h followed by puromycin selection (5 μg/ml). For transient transfections, cells were transfected with siRNA (100 nM) using Lipofectamine RNAiMAX (Invitrogen) regarding to manufacturer's instructions. Human siRNA interference for silencing P4HA2 (10uM sc-92052, Santa Cruz Biotechnology), siRNA interference for silencing RASSF1A was performed using short interfering RNA oligonucleotides GACCUCUGUGGCGACUU (Eurofins MWG). siRNA for silencing YAP1 was performed using short interfering RNA oligonucleotides CUGGUCAGAGAUACUUCUUtt (Eurofins MWG). For non-targeting control, siRNA was used with a sequence UAAGGUAUGAAGAGAUAC (Dharmacon).

### Immunofluorescence staining in 3D collagen

H1299 multicellular tumour spheroids were generated by hanging-drop method (Foty, 2011). In brief, H1299TetON-pcDNA3 and H1299TetON-RASSF1A cells were induced with 1 μg/ml doxycycline for 24 h before spheroids formation with or without 4 μM 1.4-DPCA. Cells were detached with 2 mM EDTA and re-suspended in medium supplemented with methylcellulose (20%, Sigma-Aldrich) and Matrigel matrix (1%, Corning, Growth factor reduced) and incubated as hanging droplets (25 μl) containing 2,000 cells for 48 h to generate multicellular aggregates. H1299 spheroids were washed with medium and mixed with rat tail collagen I (Serva, 2 mg/ml), 10× PBS, 1 M NaOH and complete medium. Spheroids–collagen solution was pipetted as a 100 μl drop-matrix suspension, polymerized at 37°C and replaced with medium. For 3D spheroids staining, collagen–spheroids gels were washed with PBS, fixed with 4% PFA and crosslinked in sodium azide solution overnight. Spheroids were incubated in primary antibodies (Nanog, 4903S, Cell Signaling; β-catenin, sc-376959, Santa Cruz Biotechnology; YAP, sc-101199, Santa Cruz Biotechnology; TTF-1, MA5-13961, Thermo Scientific; Mucin 5B, sc-20119, Santa Cruz Biotechnology, HIF-1α, ab51608, Abcam; all diluted 1: 100) diluted in Triton and 10% NGS overnight and after extensive washing, incubated in secondary antibodies (Alexa Fluor-488, A12379, Alexa Fluor-594, A10103, Thermo Fisher Scientific, dilution 1:1,000) overnight in 4°C. Spheroids–collagen gels were washed with PBS and mounted with DAPI. For single cells immunofluorescence staining in three-dimensional collagen, cells were trypsinized, washed in complete medium, counted and ($10^5$/ml) cells were mixed with solution containing 1.5; 2; 2.5; or 3 mg/ml collagen rat tail I (Serva), 10× PBS, 1M NaOH and complete medium. The suspension of cells–gel solution was loaded into 8 wells

(Labtech). The gels were polymerized at 37°C for 30 min and replaced with complete medium. After 48 h, the cells in 3D collagen were stained by protocol described above. Images were captured by using Nikon 20×/0.30 Ph1 objectives.

## Three-dimensional Matrigel differentiation assay

H1299 multicellular tumour spheroids were generated by hanging drops methods described above. Both H1299TetON-pcDNA3 and H1299TetON-RASSF1A multicellular aggregates were washed and cultivated on Matrigel matrix (8 mg/ml, Corning, Growth factor reduced) for 24 h. Images were monitored at 37°C using a motorized inverted Nikon Ti microscope (4×/0.10 NA air objective lens) connective to Nikon camera and captured every 30 min.

## Matrigel and collagen 3D invasion assays

RASSF1A expression was induced in H1299 with 1 μg/μl doxycycline 24 h invasion. After 24 h, cells were trypsinized and ($1 \times 10^5$) were cultured in serum-free medium in the upper wells (in triplicate) of transwell Matrigel chambers (Growth factor reduced, Corning) or transwell inserts, coated with thin layer of collagen (2 mg/ml) and allowed to invade towards bottom wells supplemented with 10% FBS conditional media. After 24 h of incubation, invading cells were fixed and stained with Richard-Allan Scientific™ Three-Step Stain (Thermo Fisher Scientific), photographed, and counted manually using Adobe Photoshop software.

## Quantitative real-time PCR analysis

RNA extraction, reverse transcription and qPCR reaction were implemented by using the Ambion® Power SYBR® Green Cells-to-CT™ kit following manufacturer's instructions (Thermo) in a 7500 FAST Real-Time PCR thermocycler with v2.0.5 software (Applied Biosystems). Calculation of mRNA fold change was analysed by using a $2^{(\Delta\Delta Ct)}$ method in relation to the YAP, P4HA2 or GAPDH reference genes.
YAP1 sense: 5′-TAGCCCTGCGTAGCCAGTTA-3′, antisense: 5′-TCAT GCTTAGTCCACTGTCTGT-3′;
GAPDH sense: 5′-TGCACCACCAACTGCTTAGC-3′, antisense: 5′-GG CATGGACTGTGGTCATGAG-3′;
P4HA2 sense: 5′-GCCTGCGCTGGAGGACCTTG-3′, antisense: 5′-TGT GCCTGGGTCCAGCCTGT-3′;
OCT4 sense: 5′-TCAGGTTGGACTGGGCCTAGT-3′, antisense: 5′-GG AGGTTCCCTCTGAGTTGCTT-3′;
SOX2 sense: 5′- GAGGGCTGGACTGCGAACT-3′, antisense: 5′- TTTG CACCCCTCCCAATTC- 3′;
NANOG sense: 5-GAAATCCCTTCCCTCGCCATC-3′and antisense: 5′-CTCAGTAGCAGACCCTTGTAAGC-3′.

## Immunoblotting

For protein analysis on 2D, cells were cultivated on 100-mm dishes and lysed in RIPA buffer as described previously (Palakurthy *et al*, 2009). For protein analyses from 3D collagen matrix, H1299 cells expressing either control pcDNA3 or RASSF1A vector were embedded into 3D collagen type I matrix. After 72 h, cells were washed two times with cold PBS and isolated by incubation in collagenase B for 10 min at 37°C, centrifuged and pellets were lysed in RIPA buffer.

Both 2D and 3D lysates were cleared by centrifugation at 22,000 ×*g* for 20 min, and protein concentration was determined by using the BSA assay, diluted in 2× NUPage sample buffer containing DTT and incubated at 99°C for 10 min. Proteins (40 μg/lane) were separated on a 10% polyacrylamide gel by SDS–PAGE and transferred to PDVF membrane (Immobilon-P, Millipore). Non-specific activity was blocked by 1×TBS containing 0.05% Tween and 4% bovine serum albumin. Membranes were probed with primary antibodies specific for TTF-1 (MA-13961, Thermo Scientific, dilution 1:500), Mucin 5B (sc-20119, Santa Cruz Biotechnology, dilution 1:500), RASSF1A (sc-58470, Santa Cruz Biotechnology, dilution 1:200), β-catenin (Santa Cruz Biotechnology, dilution 1:500), YAP (4912, Cell Signaling, dilution 1:1,000), collagen I (NB600-4080, Novusbio, dilution 1:1,000), GAPDH (97166S, Cell Signaling, dilution 1:1,000), pLATs (8654S, Cell Signaling, dilution 1:500), pYAP (4911S, Cell Signaling, dilution 1:500), LATS1 (sc-9388, Santa Cruz Biotechnology, dilution 1:200), P4HA2 (ab70887, Abcam, dilution 1:500), Nanog (4893S, Cell Signaling, dilution 1:1,000), Oct4 (2840S, Cell Signaling, dilution 1:500) and Sox2 (3579S, Cell Signaling, dilution 1:500). Membranes were then incubated with the appropriate HRP-conjugated secondary antibodies (Santa Cruz Biotechnology, dilution 1:5,000) for 1 h at room temperature. After extensive washing by TBST, the blots were developed by enhanced chemiluminescence (Millipore) and exposed by using X-Ray films.

## Immunohistochemistry

Murine lungs were collected, fixed with formalin and embedded in paraffin. Histological sections were deparaffinized, hydrated and exposed to epitope antigen retrieval with ER solution (DAKO, pH6), followed by endogenous peroxidase activity blocking for 5 min. After protein blocking (DAKO) for 60 min, sections were stained with primary antibodies, using a previously optimized dilution Nanog (4903S, Cell Signaling, dilution 1:100), YAP (4912, Cell Signaling, dilution 1:100), P4HA2 (sc-161146, Santa Cruz Biotechnology, dilution 1:50), TTF-1 (MA5-13962, Thermo Scientific, dilution 1:100), Mucin 5B (sc-20119, Santa Cruz Biotechnology, dilution 1:50) overnight in 4C. After washing, sections were incubated in secondary antibodies using EnVision detection kit (DAKO) for 30 min in room temperatures, rinsed counterstain with haematoxylin and mounted on glass slides. Slides were imaged by using Aperio ScanScope CS slide scanner, and evaluated for and percentage (0–100%) of nuclear intensity score (0, none; 1, weak; 2, moderate; and 3, strong) by using ImageScope software (Aperio).

## Animal experiments

All animal experiments were performed after local ethical committee review under a project licence issued by the UK Home Office. Lung orthotopic xenografts were generated following a published protocol (Onn *et al*, 2003) with minor modification. In brief, BALB/c nude mice (Charles River Laboratories, U.K) were anaesthetized with 2% isoflurane and placed in the right lateral decubitus position. $10^6$ of stably transfected either pcDNA3 (control)- or RASSF1A-expressing H1299, or HOP92[sccontrol] or HOP92[shRASSF1A] cells in 50 μl of 50% Matrigel (BD Biosciences) were injected into the left lung. Mice were sacrificed on day 17 or day 30, and collected lungs were analysed for their tumours.

## Immunofluorescence microscopy

Cells were seeded and cultivated either on coverslips or on the collagen coated wells with defined stiffness of collagen matrix (soft slips 12, SS12-COL-0.5, SS12-COL-4, SS12-COL-25), fixed in 4% paraformaldehyde for 15 min at room temperature and then permeabilized with 0.5% Triton X-100 (Sigma) in PBS for 10 min at room temperature. Non-specific binding was blocked with 3% bovine serum albumin (Sigma) in PBS for 30 min before incubation with a primary antibody against type I collagen (NB-600-4080, Novusbio, dilution 1:100), Nanog (4903S, Cell Signaling, dilution 1:100), YAP (sc-101199, Santa Cruz Biotechnology, dilution 1:100) and P4HA2 (sc-161146, Santa Cruz Biotechnology, dilution 1:100) for 2 h at room temperature. Secondary antibodies (Alexa conjugated-488 and 594, Thermo Fisher Scientific) were applied for 1 h at room temperature, followed by staining with or without Phalloidin (Life Technologies) for 15 min, and after extensive washing between each step, coverslips were mounted onto microscopy slides with mounting medium containing DAPI. Images were captured by using confocal Nikon 60×/1.25 objectives. For each condition, a minimum of 300 cells were analysed.

## Collagen contraction assay

Collagen gels for contraction assays were prepared by using of Rat tail collagen I (Serva, Germany), 10× DMEM, 1 M NaOH (final pH 7.4) and mixed with cells $(1 \times 10^6/ml)$ in 8:1:1:1 ratio previously described in Kamel *et al* (2014), with final concentration 2 mg/ml. Gel–cells suspension (triplicates) was pipetted into 96 wells precoated with BSA and allowed polymerized in 5% $CO_2$ at 37°C for at least 30 min. The collagen lattices containing cells were replaced with completed medium with or without inhibitor and allowed to contraction at least 4 days. After day 4, the images were taken and contraction was calculated as a decreased area of original diameters area of 96 wells. Comparison of collagen gel contraction was performed by using Student's unpaired one-tail *t*-test, and $P < 0.05$ were considered statistically significant.

## Hydroxyproline assay

Measurement of hydroxyproline ratio was provided by Hydroxyproline Assay Kit (perchlorate-free) (Cell Biolabs, Inc.) according to manufacturer's instruction. Shortly, $3–6 \times 10^6$ H1299 cells were collected and incubated with 12N hydrochloric acid and hydrolysed for 3 h at 120°C. After brief cooling, 5 mg of activated charcoal was added, properly mixed and centrifuged for 5 min at $10,000 \times g$. Acid-hydrolysed samples were recovered and evaporated under vacuum at 60–80°C for 45 min. Samples were incubated with Chloramine T for 30 min, followed by Ehrlich reagent for another 45 min at 60°C. Samples were centrifuged at $6,000 \times g$ for 15 min, and absorbance of each sample was measured on microplate reader using 540–560 nm as the primary wavelength.

## Picrosirius red staining

4 μm paraffin sections were collected on 3-aminopropyltriethoxysilane (AAS) slides, and staining was performed in accordance with manufacturer's protocol (Abcam, ab150681). Briefly, slides were deparaffinized in graded ethanol solutions and then placed in distilled water for 5 min. Picrosirius red solution was placed on slides for 60 min at RT and then rinsed quickly with acetic acid solution. After staining, slides were passed through graded ethanol solutions, cleared in acetone and mounted with synthetic resin.

## Generation of cell-free extracellular matrix

$2 \times 10^5$ stably transfected H1299 cells either with pcDNA3 H1299$^{control}$ or H1299$^{RASSF1A}$ vector were seeded on coverslips in 12 wells to allow them to produce extracellular matrix for 7 days with addition of 100 μg/ml L-ascorbic acid and with or without silencing with siRNA against P4HA2 or treatment with P4HA inhibitor. Coverslips were washed with PBS, and ECM was extracted five times for 5 min at 4°C with 0.5% DOC in immunoprecipitation buffer under gentle shaking how was previously described (Unsöld *et al*, 2001). After discarding cell debris, ECM was washed three times with PBS to clean ECM of any remaining debris. $5 \times 10^4$ H1299 RASSF1A cells were seeded on cell-free extracellular matrix extracted from either control or RASSF1A cells and grown for 24 h to see effect of ECM on YAP expression and localization. Both H1299$^{control}$ and H1299$^{RASSF1A}$ cells were fixed and immunofluorescently stained with YAP antibody (sc-101199, Santa Cruz Biotechnology, dilution 1:100).

## Mass spectrometry

H1299 cells expressing empty pcDNA3 or RASSF1A vector were growing for 2 weeks in the presence of 100 μg/ml L-ascorbic acid. ECM was extracted by 0.5% DOC (Unsöld *et al*, 2001) and incubated with 0.5 M acetic solution overnight at 4°C. Collected ECM in acetic acid was reduced by 20 mM DTT (Sigma), followed by incubation in 30 mM iodoacetamide alkylating reagent (Sigma). Proteins from ECM were precipitated via methanol/chloroform extraction and re-suspended in 6 M urea in 0.1 M Tris pH7.8 by vortexing and sonication. Final protein concentration was measured by BSA. Thirty micrograms of protein was further diluted to bring urea < 1 M and digested with immobilized trypsin (Fisher Scientific) overnight at 37°C. Digestion was stopped by acidification of the solution with 1% trifluoroacetic acid (TFA, Fisher Scientific). After digestion, samples were desalted by solid phase using C18+carbon Spin tip (Ltd) and dried down using a Speed Vac. Dried tryptic peptides were re-constituted in 20 μl of 2% acetonitrile-98% $H_2O$ and 0.1% TFA and subsequently analysed by nano-LC LS/MS using a Dionex Ultimate 3000 UPLC coupled to a Q Exactive HF mass spectrometer (Thermo Fisher Scientific) (Vaz *et al*, 2016). LC-MS/MS data were searched against the Human UniProt database (November 2015; containing 20274 human sequences) using Mascot data search engine (v2.3). The search was carried out by enabling the Decoy function, whilst selecting trypsin as enzyme (allowing 1 missed cleavage), peptide charge of +2, +3, +4 ions, peptide tolerance of 10 ppm and MS/MS of 0.05 Da; #13C at 1; carbamidomethyl (C) as fixed modification; and oxidation (M), deamidation (NQ) and phosphorylation (STY) as a variable modification. MASCOT data search results were filtered using ion score cutoff at 20 and a false discovery rate (FDR) of 1%. A qualitative analysis of proteins identified was performed using a Venn diagram, and only proteins identified with two or more peptides were taken into consideration. The resulting list of proteins was culled to ECM proteins.

## Magnetic resonance imaging

Mouse MRI was performed on day 30 after H1299 cell implantation using a 4.7 Tesla, 310 mm horizontal bore magnet equipped with a 120 mm bore gradient insert capable of 400 milliTesla/metre (mT/ m) in all three axes (Varian Inc, CA). RF transmission and reception was performed with a 30-mm-long, 25-mm quadrature birdcage coil (Rapid Biomedical GmbH, Germany). Balanced steady-state free precession (SSFP) scans were acquired (repetition time (TR) = 2.684 ms, echo time (TE) = 1.342 ms, flip angle (FA) = 20°, field of view (FOV) = $48 \times 24 \times 24$ mm$^3$, matrix = $256 \times 96 \times 96$ and RF hard pulse duration = 16 μs). MR images were analysed using ITK-SNAP (Yushkevich *et al*, 2006).

## Mechanical analyses by scanning probe microscopy

5 μm slides from frozen lung tissue were transferred on coverslips and analysed for stiffness of primary tumours by AFM. For *in vitro* three-dimensional stiffness analyses, $2 \times 10^5$ H1299 cells expressing either pcDNA3 H1299$^{control}$ or re-expressing RASSF1A H1299$^{RASSF1A}$ were embedded into 3D rat tail collagen type I matrix and allowed them modified ECM for 5 days in 8 wells (Labtech). 3D collagen gels were transferred onto slides and subjected to mechanical testing by scanning probe microscopy. Scanning probe microscopy was performed on a MFP-3D Atomic Force Microscope (Asylum Research, High Wycombe, UK), with an AC240TS probe ($k = 2.0$ Nm$^{-1}$, Olympus, Japan). AMFM nanomechanical mapping and loss tangent imaging (Proksch & Yablon, 2012) were applied, with measurements based on the shift in the probe's resonant frequencies dependent on the strength of the interaction, or tip-sample contact force (Giessibl, 1997).

A tip correction factor was calculated based on the known compressive Young's modulus of a polycaprolactone calibration sample (300 MPa). Three $5 \times 5$ μm areas were then randomly selected and measured from each material. $512 \times 512$ pixel maps of height, Young's modulus and loss tangent were recorded for each image.

## Second harmonic generation using high resolution microscopy

Second harmonic generation signal from three-dimensional spheroids in 3D collagen matrix or from tumour tissue was detected upon simultaneous excitation with a 920-nm laser (MaiTai). The signal was collected using a BP 460/50 filter, MBS 690. The images were acquired using the LSM 7 MP microscope (Carl Zeiss, Jena, Germany) using the Zeiss W Plan-Apochromat 20×/0.8 M27 or Plan-Apochromat 63×/1.4 Oil Dic M27 objective lens.

## Statistical analysis

The results from all experiments represent the means ± SEM of replicated samples or randomly imaged cells within the field. Numerical values of cell culture and mouse cohort data were analysed using Student's *t*-test for significance in Prism 6 (GraphPad Software Inc.). Unpaired two-tailed Student's *t*-test was used to compare the mean values of two groups. The difference was considered as statistical significant of $P < 0.05$.

## TCGA survival analysis

The Cancer Genome Atlas (TCGA) project of Genomic Data Commons (GDC) collects and analyses multiple human cancer samples. The TCGA RNA-seq data were mapped using the Ensembl gene id available from TCGA, and the FPKMs (number Fragments per Kilobase of exon per Million reads) for each gene were subsequently used for quantification of expression with a detection threshold of 1 FPKM. Genes were categorized using the same classification as described above. Based on the FPKM value of each gene, patients were classified into two expression groups and the correlation between expression level and patient survival was examined. Genes with a median expression less than FPKM 1 were excluded. The prognosis of each group of patients was examined by Kaplan–Meier survival estimators, and the survival outcomes of the two groups were compared by log-rank tests. Maximally separated Kaplan–Meier plots are presented with log-rank *P* values. Survival analysis was performed using SPSS version 24.0.

# Data availability

The mass spectrometry raw data included from this publication have been deposited to the ProteomeXchange Consortium via the PRIDE partner repository and assigned the identifier PRIDE: PXD012694 (https://www.ebi.ac.uk/pride/archive/projects/PXD012694).

**Expanded View** for this article is available online.

## Acknowledgements

We would like to thank R. Fischer and B. Kessler from the TDI MS Laboratory, for Mass spectrometry. We would like to acknowledge J.M. Kurie and P. Friedl from M.D. Anderson Cancer Center, Houston, Texas, USA, for very valuable comments during writing manuscript. We also thank S. Smart, A. Gomes and D. Allen from imaging core facility for MRI analyses and graphical assistance. We thank R.Wilson and G.Brown from Microscopy core for technical support. This work was supported by Cancer Research UK A19277 and the Medical Research Council.

## Author contributions

EON and DP designed the research, performed experiments and analysed data. YJ and AR assisted with animal work, MC provided RT–PCR, IV provided and analysed *LC-MS/MS*, JB performed the fibrotic staining, CB performed AFM, and EON and DP wrote manuscript.

## Conflict of interest

The authors declare that they have no conflict of interest.

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
