## [Review Process File · The EMBO Journal]

RASSF1A controls Tissue Stiffness and Cancer Stem-like Cells in Lung Adenocarcinoma

Daniela Pankova, Yanyan Jiang, Maria Chatzifrangkeskou, Iolanda Vendrell, Jon N. Buzzelli, Anderson Ryan, Cameron Brown and Eric O'Neill

Review timeline:	Submission date:	21st Aug 2018
	Editorial correspondence:	19th Sep 2018
	Authors' correspondence:	15th Oct 2018
	Editorial Decision:	18th Oct 2018
	Revision received:	3rd Feb 2019
	Editorial Decision:	21st Mar 2019
	Revision received:	23rd Apr 2019
	Accepted:	29th Apr 2019

Editor: Daniel Klimmeck

Transaction Report:

Editorial Correspondence

19th Sep 2018

Thank you for the submission of your manuscript (EMBOJ-2018-100532) to The EMBO Journal. My apologies for the delay in getting back to you at this time of the year. Your study has been sent to three referees for evaluation, however referee #2 did not deliver his/her report even after repeated notes from our side. We have in the meantime received the reports from both referees, which I copy below. In the interest of time, we have decided to move on with our decision based on these reports.

As you will see, the referees acknowledge the potential interest and novelty of your work, although they also express major concerns. In particular, referee #1 raises reservations in that the claims on functional links between Rassf1a, rigidity and YAP are not sufficiently supported by the data in his/her view and states that the relevance of your findings and mechanistic details remain unclear. Referee #3 agrees in that the signaling details upon RASSF1A overexpression is not conclusively explored. In addition, the referees points out that the underlying causalities between stiffness levels and increased tumorigenesis would need to be conclusively addressed.

These are important points in our view, and given the substantial criticisms raised, we find it difficult to commit to going further with this manuscript The EMBO Journal.

However, before making the final decision, I would offer you the chance to read the reports and to let us know about your view on the critique and how the concerns raised by the referees could be addressed within the time-frame of a revision. It would therefore be helpful if you could already at this point provide me with a preliminary point-by-point response on what data could be included in the revised manuscript. In this way, we can better agree on the exact experimental requirements for the revision. I will then re-consult with the referees to determine if such a revision would address their concerns. I would like to stress that I need strong endorsement from the referees in order to fully commit to a revised manuscript.

Please feel free to contact me with any questions related to this matter. By conducting this exchange at the current stage I hope to avoid inviting a revision with a high risk of being rejected by the referees following extensive experimental efforts on your side.

REFeree REPORTS:

EMBOJ-2018-100532

Comments for the authors:

Referee #1:

They carry out RASSF1A overexpression in one lung adenocarcinoma cell line H1299. These cells are injected in the lung to generate primary tumors. RASSF1A overexpression has no consequence on primary tumor formation although 3/6 injected mice had less metastases when compared to parental H1299 control (con).

1) The statistical significance of these results remains unclear. There are many variables associated with a lung orthotopic injection (Pneumothorax? Lung damage? spilling contralaterally). Also unclear is whether this is an intrinsic slightly reduced metastatic potential of the H1299 clone selected to carry RASSF1A.

So the main result shown in Fig.1 is not impressive (in magnitude) and technically questionable. I would be more convinced if this were complemented by inactivation of RASSF1A in another cell lines (such as the HOP92 used in Fig3) that express it and show that this is increasing metastatic colonization in the same assay.

2) In Fig2 they provide an intriguing result. They find that Rassf1a expressing cells display less nuclear YAP in cells at an intermediate rigidity. This can have many explanations, but the simplest one is RASSF1A/Hippo signaling may be potentially downstream of - or in a negative feedback with - the cell's response to the ECM. So raising RASSF1A would be expected to blunt YAP by intercepting its activation downstream of the ECM. Instead, and this is the unexpected and interesting finding, they show (pity that this is explained imperfectly in the text...) that a different story is going on: the main culprit is not a signal transduction issue within the cell, rather it is the rigidity (or whatever "quality" of the ECM secreted by control cells, but see point below, and not from RASSF1A expressing cells) the dominant signal or main culprit able to trigger nuclear YAP independently of RASSF1A signaling levels. It seems that ECM mechanics overrules raised RASSF1A signaling, disabling its YAP inhibition, and sending YAP to the nucleus anyway. I think that this section should be extensively rewritten and properly discussed to make this understandable by the non-expert readership.

3) There is however some confusion in these experiments that needs to be addressed. How do they know that H1299 control produce stiffer matrix than the Rassf1 overexpressing one. This is not shown by the second harmonics in the lung (the source of collagen there can be CAFs), and the gel contraction speaks in favor of the differential ability of the two cells to build up a contractile response (that is more related to integrin and cytoskeleton) than to their own ECM production, at least in principle. This aspect needs some dissection and clarification. One way is to show relevance of the P4HA2 gene that is differentially expressed (in the control and rassf1 condition). See next point. For this concern, I would like to see first demonstrated that P4H42 is essential for gel contraction of H1299 control cells (and thus it is the endogenously produced collagen deposited on top of the synthetic/exogenous/experimental collagen that is responsible for the phenotype).

4) The authors are not addressing further the mechanism outlined above in fig2, but rather ask the complementary question of what RASSF1A expression does to the ECM secreted by H1299 cells that disables the positive ECM-YAP axis of control cells. They find reduced P4HA2 (potentially relevant for procollagen folding), although most of their observations on this protein in Figure 5 offers an interesting set of data, but mainly correlative, lacking functional validation. P4HA2 is a central hinge for this paper, hinting to a ECM-YAP-ECM positive feedback loop, but without causality it is hard to appreciate most of these claims. The use of a P4HA2 inhibitory drug of unclear specificity in Figure 6 is not sufficient and needs itself validation. Is depletion of P4HA2 sufficient to

make the ECM of control cells similar to that one of RASSF1A cells?

5) If they validate the P4HA2 inhibitor 1,4-DPCA, then the finding that reducing YAP-induced ECM stiffness, that in turn sustains/feedbacks on YAP-induced stemness, would start to make much more sense.

6) In this light, the paragraph "Activation of Hippo pathway leads to cancer cell differentiation" really left me struggling. This is at the end of a paper showing that it is the ECM and its properties to be ultimately the determinant of YAP activation and that RASSF1A is a mean to attenuate a positive feedback loop of YAP on the ECM. This should be titled "P4HA2-mediated collagen synthesis attenuates cancer cell differentiation"

7) In figure 7 they use established markers for lung adenocarcinoma progression studies, that is mucin and TTF1. however in previous figures they use Nanog positivity as immediate read-out of YAP activity and ECM rigidity. how established is that?

8) the claim on Wnt signaling is a dramatic detour that goes nowhere and should be deleted. " As expected, We interpret this data, together with the isolation of 5T4/TPBG in H1299con ECM to suggest that reduced binding of 5T4/TPBG to the ECM upon P4HA2i allows 5T4/TPBG to suppress WNT signaling and destabilize b-catenin"

This is correlative and premature and I have no way to know where is this coming from. Alternative there should be some experimental validation in support of a role of Wnt signaling (note that intrinsic suppression is really hard to follow)

9) More generally they are not citing key and supporting references in the result or discussion section that makes the reading very fragmented (I kept asking myself: have they shown that? and where...are they relying on some other data published by others? this is impossible to follow unless one is a really super-expert).

10) the title of the paper is misleading. There is very little Hippo signaling investigated here and the connections of RASSF1A with hippo kinases is not thoroughly established in lung tumorigenesis. Thus, a more appropriate title should be
RASSF1A controls Tissue Stiffness and Cancer Stem-like Cells in Lung Adenocarcinoma

In sum, the MS is interesting but with important gaps. They should remove and make sure, and restructure the Figure in more logical manner, rather than add data on different directions. Their interpretational lines are often questionable and should be streamlined.

Referee #2:

In the paper by Pankova et al, the authors describe a novel mechanism by which loss of RASSF1A induce YAP signaling and induction of tissue stiffness through upregulation of the P4HA2 collagen modifying enzyme. The authors suggest that loss of RASSF1A, a tumour suppressor, and the consequent loss of hippo signaling induces stiffness-dependent beta-catenin signaling that drives cancer stemness and metastatic progression of lung adenocarcinomas but not squamous carcinomas. The story is interesting, and potentially true, but the quality of the data are often of very low standard and it is difficult to interpret and conclude much of their data. Most of the imaging are of very poor quality and not presenting themselves from a convincing side. The text is poorly written and many typos and mistakes are presented in the text as well as in the figure legends. In general, the data are presented in a confusing manner and no flow is obtained when reading the paper. This reviewer strongly suggests the authors of re-visit the order of the presented data in order to generate a nice flow.

The paper in its current form is not of sufficient quality to be published in EMBO Journal. However, with a substantial revision, the study might still reach scientific interest for the readers of EMBO Journal. This reviewer would give the authors the benefit of the doubt if a substantial effort is performed.

Major Comments:

- A much better characterisation of the cells after overexpression of RASSF1A. Could the authors please show expression levels of the following: pMST1/2, pLATS1/2, pS127-YAP, mRNA levels of a few YAP-target genes and cell proliferation data/curves.
- The data in Figure 1E representing the metastatic disease are very confusing and surprising for several reasons. First, why are there all 6 control mice having contralateral metastasis but only 5 of the control mice actually have a detectable primary tumours (Fig 1D)? Second, how come only 4 of the 6 control mice have local ipsilateral mets but all 6 have distant contralateral mets? Please try to explain why some of the mice do not have local mets but only distant mets. Finally, how was the number of metastasis actually determined/quantified? Could there be some problems with the method?
- The in vitro transwell assays of cell invasion should also be conducted through collagen-1. First of all, collagen seem to be a far more important contributor to the molecular mechanism presented in the study (although Lamin B2 is upregulated in their mass spec data). Second of all, invasion through matrigel mimics invasion through basement membranes while invasion through collagen mimic invasion through the parenchyma tissue.
- One major issue through the paper is whether tissue stiffness or the actual collagen concentration is to be responsible for their observations. When using higher conc. of collagen it is true that the stiffness increases but also the available epitopes (avidity) for cell binding. It is therefore very important to validate what is driving the progression in their study. This reviewer demand to see experimental set-ups, which discriminate between stiffness and collagen concentration. One way of validating stiffness vs collagen conc. is to use polyacrylamid gels of different stiffness and then coat them with the same collagen conc. One can even purchase custom made gels from Matrigel (see softwells). This reviewer is in doubt if it is actually stiffness and not collagen avidity that drives Nanog expression. Indeed, Fig EV3D clearly demonstrate that cell plated on 2D glass, which is much stiffer than 3% (3 mg/ml) collagen actually have lower levels of Nanog in the nucleus!!!!
- A second major problem with this paper is that the mechanism is proposed to go through P4HA2. But only one experiment using a P4HA2 inhibitor is used to rescue the effect of Nanog expression. No validation of the inhibitors specificity is shown, and no other rescue experiments have been conducted. This reviewer demands that rescue experiments are conducted using ablation of P4HA2 in cells lacking RASSF1A expression. These should include, gel contraction, collage production, ECM stiffness, beta-catenin, stemness (Nanog, Oct4, SOX expression) and potentially also include in vivo validation (although the in vivo part may seem harsh to demand).
- The experimental data on 5T4 proteins are extremely weak. I think the data does not allow any kind of conclusions without further experimentation.
- In the model in Figure 8, the author mention OCT4 and SOX2, but no data is presented in the paper! Could they include data on these two genes as well as Nanog?
- The authors show that control cells produce linearized collagen bundles emanating from the spheroid in Fig 4A. As the spheroids are made within collagen matrices this is not so surprising. In fact, one should even detect collagen bundles like that in gels without any cells. So, if the conclusion is that more bundles are produced or formed by the cells not expressing RASSF1A, then the authors need to do more work to convince the reader. I.e. many spheroids have to be quantified from both control cells and RASFF1A expressing cells, and these have to be compared to gels without any cells inside. And please recall that the presence of matrigel also affect collagen bundling.
- Is the orthotopic implantation inducing inflammation in the lung, that could drive collagen remodeling per se. Could they authors i.e. perform a tail-vein injection to prevent inflammation in the lung and recapitulate the data?

Minor Comments:

- All antibodies used in the paper as to be specified with catalog numbers.
- Several places the authors describe the collagen conc. as percentage. Please correct this to mg/ml. There is a huge difference between 3% and 3 mg/ml collagen.
- Some lines are repeated in the M&M under the 'immunoblotting' paragraph
- In the 'Three-dimensional Matrigel migration assay' paragraph: which percentage of Matrigel is use?
- In the 'Immunofluorescence staining in 3D collagen' paragraph: 20% methyl cellulose seems very high. Is this correct? Also, please specific the conc. of TX100 used for permeabilisation during the incubation step.

- In the 'Collagen contraction assay' paragraph: the authors write that the use 8 part collagen-1. Could they specific the concentration of collagen-1 instead?
 - o The authors state that an increase in gel diameter was used to calculate the contraction. The authors obviously mean the decrease in gel diameter. Please correct.
 - o Just a helping suggestion: instead of releasing the gel from the plastic well using a needle, it is much easier to pre-coat the wells with 1%BSA, PBS for 1-2 hours and then plate the gels within the well. BSA-coating prevent gel attachment to the plastic
- It is impossible to see the actual gels in the contraction assay in Fig 2B. Please correct.
 - o The quantification of the 24 well size gel contracted is estimated to be between 1-2 mm³. This is nothing, something must be wrong and should be corrected.
- In general, the SHG images are of very poor quality with exception of Fig 4C.
- Fig 2C: they show SHG of the ipsilateral metastasis at 17days but in Fig1 and in the text they state no metastasis is present at day 17. So, why are they showing this images? The authors are not claiming that they look for pre-metastatic niches - or are they? This reviewer does not understand what this images is telling the reader?
- There is extremely little YAP in the nucleus also in the control cells in Fig 2D. It is hard to judge if the images have been taken in the same z-plane or if the YAP staining is actually not in the cytoplasm under or above the nucleus? Poor quality again.
- The authors claim to have quantified YAP nuclear translocation in Fig 2E. But how do they do this - no description is found in the method section? In addition, it would be more useful to quantify the nuclear/cytoplasmic ratio of individual cells.
- The immunofluorescence detection of P4HA2 is very weak in Fig 3C&D. Are they sure the antibody is detecting the protein by IF. Could the authors provide evidence of this staining after P4HA2 depletion, and could they provide a negative control for the P4HA2 staining in general?
- Iis the quantification of collagen derived from the primary tumour or a metastasis in Figure 4C?
 - o Why are they showing a single SHG image of a control-metastasis but not a RASSF1A-metastasis? What do they want to tell with that? Either the authors need to show both cell types or they should remove this image.
- In Fig 5E the graph states fibrotic area. Please simply state that you quantified Picrosirius Red are rather than 'fibrosis area'. Be specific.
- Something is wrong with the display of the images in Figure 6B. For instance, column 2 has one 'green-colored' images while the others are blue?
 - o There are two rows of Merge. What is actually merged here? Please specify.
- Again what is merged in Fig 6F? Dapi and Nanog? Please specify. Importantly these images in Fig 6F seem to be out of the plane, as there is not even a DAPI signal in some of the images (forth row). The Fig 6F would also benefit of some translocation quantification (ratio: nuclear/cytoplasmic signal).
- Why do the authors observe a difference in ipsilateral and contralateral SHG area (Fig 2C and EV2C)?
- What is on the x-axis of Fig EV3D?
- In Fig EV3A: 0.6 or 60% of the Nanog signal does not seem to correspond to the stainings - no way that 60% is in the nucleus when looking at the images!?! This quantification has to be re-evaluated. It seems way to high.
- What do the authors mean by stable collagen (first line page 6)?
- Second line page 7. The authors write 1.1 kPa but the graph shows 11 kPa. Please correct.
- Fig 2C. How did they quantify the collagen volume? How did they end up with 2-8 volume%? Should it not be area at least?

Authors' correspondence

15th Oct 2018

Many thanks for the opportunity to preliminarily address the reviewers concerns regarding our original submission. We are grateful for the opportunity and feel the comments were extremely helpful and readily addressable. We have outlined the responses and proposed experiments suggest which we agree will clarify and strengthen the story.

In light of the general positive comments from both reviewers regarding the underlying story, we hope you find this plan acceptable and will be willing to accept a revised manuscript in due course.

Thank you again for the submission of your manuscript (EMBOJ-2018-100532) to The EMBO Journal and in addition providing us with a preliminary revision plan. Thank you also for your patience with my response, which got delayed due to detailed discussions in the team regarding your preliminary point-by-point response. As mentioned earlier, your study has been sent to two referees, and we received reports from both of them, which I enclose below.

The referees acknowledge the potential interest and novelty of your work, although they also express major concerns. In particular, referee #1 raises reservations in that the claims on functional links between RASSF1a, rigidity and YAP are not sufficiently supported by the data in his/her view and states that the relevance of your findings and mechanistic details remain unclear. Referee #3 agrees in that the signaling details upon RASSF1A overexpression is not conclusively explored. In addition, the referees point out that the underlying causalities between stiffness levels and increased tumorigenesis would need to be conclusively addressed. In addition, the referees point to issues related to terminology, experimental design, documentation of methodologies and statistics that would need to be conclusively addressed to achieve the level of robustness needed for The EMBO Journal.

We realise that you would - judging from the information provided in the point-by-point letter - be potentially able to address the issues raised by the referees in a revised version of the manuscript.

I judge the comments of the referees to be generally reasonable and can - based on your sensible preliminary response - offer to invite you to revise your manuscript experimentally to address the referees' concerns. I agree that in particular the aspect of more rigorous support for the connection between RASSF1a, rigidity and YAP, involvement of P4HA2, and the distinction between stiffness and collagen would need to be conclusively addressed in a revised version of the manuscript to move towards publication.

Please feel free to contact me if you have any questions or need further input on the referee comments.

Referee #1:

They carry out RASSF1A overexpression in one lung adenocarcinoma cell line H1299. These cells are injected in the lung to generate primary tumors. RASSF1A overexpression has no consequence on primary tumor formation although 3/6 injected mice had less metastases when compared to parental H1299 control (con).

1) The statistical significance of these results remains unclear. There are many variables associated to a lung orthotopic injection (Pneumothorax? Lung damage? spilling contralaterally). Also unclear is whether this is an intrinsic slightly reduced metastatic potential of the H1299 clone selected to carry RASSF1A.

So the main result shown in Fig.1 is not impressive (in magnitude) and technically questionable. I would be more convinced if this were complemented by inactivation of RASSF1A in another cell lines (such as the HOP92 used in Fig3) that express it and show that this is increasing metastatic colonization in the same assay.

2) In Fig2 they provide an intriguing result. They find that RASSF1a expressing cells display less nuclear YAP in cells at an intermediate rigidity. This can have many explanations, but the simplest one is RASSF1A/Hippo signaling may be potentially downstream of - or in a negative feedback with - the cell's response to the ECM. So raising RASSF1A would be expected to blunt YAP by intercepting its activation downstream of the ECM. Instead, and this is the unexpected and interesting finding, they show (pity that this explained imperfectly in the text...) that a different story is going on: the main culprit is not a signal transduction issue within the cell, rather it is the rigidity (or whatever "quality" of the ECM secreted by control cells, but see point below, and not from RASSF1A expressing cells) the dominant signal or main culprit able to trigger nuclear YAP

independently of RASSF1A signaling levels. It seems that ECM mechanics overrules raised RASSF1A signaling, disabling its YAP inhibition, and sending YAP to the nucleus anyway. I think that this section should be extensively rewritten and properly discussed to make this understandable by the non-expert readership.

3) There is however some confusion in these experiments that needs to be addressed. How do they know that H1299 control produce stiffer matrix than the *Rassf1* overexpressing one. This is not shown by the second harmonics in the lung (the source of collagen there can be CAFs), and the gel contraction speaks in favor of the differential ability of the two cells to build up a contractile response (that is more related to integrin and cytoskeleton) than to their own ECM production, at least in principle. This aspect needs some dissection and clarification. One way is to show relevance of the P4HA2 gene that is differentially expressed (in the control and *rassf1* condition). See next point. For this concern, I would like to see first demonstrated that P4HA2 is essential for gel contraction of H1299 control cells (and thus it is the endogenously produced collagen deposited on top of the synthetic/exogenous/experimental collagen that is responsible for the phenotype).

4) The authors are not addressing further the mechanism outlined above in fig2, but rather ask the complementary question of what RASSF1A expression does to the ECM secreted by H1299 cells that disables the positive ECM-YAP axis of control cells. They find reduced P4HA2 (potentially relevant for procollagen folding), although most of their observations on this protein in Figure 5 offers an interesting set of data, but mainly correlative, lacking functional validation. P4HA2 is a central hinge for this paper, hinting to a ECM-YAP-ECM positive feedback loop, but without causality it is hard to appreciate most of these claims. The use of a P4HA2 inhibitory drug of unclear specificity in Figure 6 is not sufficient and need itself validation. Is depletion of P4HA2 sufficient to make the ECM of control cells similar to that one of RASSF1A cells?

5) If they validate the P4HA2 inhibitor 1,4-DPCA, then the finding that reducing YAP-induced ECM stiffness, that in turn sustains/feedbacks on YAP-induced stemness, would start to make much more sense.

6) In this light, the paragraph "Activation of Hippo pathway leads to cancer cell differentiation" really left me struggling. This is at the end of a paper showing that it is the ECM and its properties to be ultimately the determinant of YAP activation and that RASSF1A is a mean to attenuate a positive feedback loop of YAP on the ECM. This should be titled "P4HA2-mediated collagen synthesis attenuates cancer cell differentiation"

7) In figure 7 they use established markers for lung adenocarcinoma progression studies, that is mucin and TTF1. however in previous figures they use Nanog positivity as immediate read-out of YAP activity and ECM rigidity. how established is that?

8) the claim on Wnt signaling is a dramatic detour that goes nowhere and should be deleted. " As expected, We interpret this data, together with the isolation of 5T4/TPBG in H1299con ECM to suggest that reduced binding of 5T4/TPBG to the ECM upon P4HA2i allows 5T4/TPBG to suppress WNT signaling and destabilize b-catenin"

This is correlative and premature and I have no way to know where is this coming from. Alternative there should be some experimental validation in support of a role of Wnt signaling (note that intrinsic suppression is really hard to follow)

9) More generally they are not citing key and supporting references in the result or discussion section that makes the reading very fragmented (I kept asking myself: have they shown that? and where...are they relying on some other data published by others? this is impossible to follow unless one is a really super-expert).

10) the title of the paper is misleading. There is very little Hippo signaling investigated here and the connections of RASSF1A with hippo kinases is not thoroughly established in lung tumorigenesis. Thus, a more appropriate title should be
RASSF1A controls Tissue Stiffness and Cancer Stem-like Cells in Lung Adenocarcinoma

In sum, the MS is interesting but with important gaps. They should remove and make sure, and restructure the Figure in more logical manner, rather than add data on different directions. Their interpretational lines are often questionable and should be streamlined.

Referee #3:

In the paper by Pankova et al, the authors describe a novel mechanism by which loss of RASSF1A induce YAP signaling and induction of tissue stiffness through upregulation of the P4HA2 collagen modifying enzyme. The authors suggest that loss of RASSF1A, a tumour suppressor, and the consequent loss of hippo signaling induces stiffness-dependent beta-catenin signaling that drives cancer stemness and metastatic progression of lung adenocarcinomas but not squamous carcinomas. The story is interesting, and potentially true, but the quality of the data are often of very low standard and it is difficult to interpret and conclude much of their data. Most of the imaging are of very poor quality and not presenting themselves from a convincing side. The text is poorly written and many typos and mistakes are presented in the text as well as in the figure legends. In general, the data are presented in a confusing manner and no flow is obtained when reading the paper. This reviewer strongly suggests the authors of re-visit the order of the presented data in order to generate a nice flow.

The paper in its current form is not of sufficient quality to be published in EMBO Journal. However, with a substantial revision, the study might still reach scientific interest for the readers of EMBO Journal. This reviewer would give the authors the benefit of the doubt if a substantial effort is performed.

Major Comments:

- A much better characterisation of the cells after overexpression of RASSF1A. Could the authors please show expression levels of the following: pMST1/2, pLATS1/2, pS127-YAP, mRNA levels of a few YAP-target genes and cell proliferation data/curves.
- The data in Figure 1E representing the metastatic disease are very confusing and surprising for several reasons. First, why are there all 6 control mice having contralateral metastasis but only 5 of the control mice actually have a detectable primary tumours (Fig 1D)? Second, how come only 4 of the 6 control mice have local ipsilateral mets but all 6 have distant contralateral mets? Please try to explain why some of the mice do not have local mets but only distant mets. Finally, how was the number of metastasis actually determined/quantified? Could there be some problems with the method?
- The in vitro transwell assays of cell invasion should also be conducted through collagen-1. First of all, collagen seem to be a far more important contributor to the molecular mechanism presented in the study (although Lamin B2 is upregulated in their mass spec data). Second of all, invasion through matrigel mimics invasion through basement membranes while invasion through collagen mimic invasion through the parenchyma tissue.
- One major issue through the paper is whether tissue stiffness or the actual collagen concentration is to be responsible for their observations. When using higher conc. of collagen it is true that the stiffness increases but also the available epitopes (avidity) for cell binding. It is therefore very important to validate what is driving the progression in their study. This reviewer demand to see experimental set-ups, which discriminate between stiffness and collagen concentration. One way of validating stiffness vs collagen conc. is to use polyacrylamid gels of different stiffness and then coat them with the same collagen conc. One can even purchase custom made gels from Matrigen (see softwells). This reviewer is in doubt if it is actually stiffness and not collagen avidity that drives Nanog expression. Indeed, Fig EV3D clearly demonstrate that cell plated on 2D glass, which is much stiffer than 3% (3 mg/ml) collagen actually have lower levels of Nanog in the nucleus!!!!
- A second major problem with this paper is that the mechanism is proposed to go through P4HA2. But only one experiment using a P4HA2 inhibitor is used to rescue the effect of Nanog expression. No validation of the inhibitors specificity is shown, and no other rescue experiments have been conducted. This reviewer demands that rescue experiments are conducted using ablation of P4HA2 in cells lacking RASSF1A expression. These should include, gel contraction, collage production, ECM stiffness, beta-catenin, stemness (Nanog, Oct4, SOX expression) and potentially also include in vivo validation (although the in vivo part may seem harsh to demand).
- The experimental data on 5T4 proteins are extremely weak. I think the data does not allow any kind of conclusions without further experimentation.
- In the model in Figure 8, the author mention OCT4 and SOX2, but no data is presented in the paper! Could they include data on these two genes as well as Nanog?
- The authors show that control cells produce linearized collagen bundles emanating from the spheroid in Fig 4A. As the spheroids are made within collagen matrices this is not so surprising. In fact, one should even detect collagen bundles like that in gels without any cells. So, if the conclusion is that more bundles are produced or formed by the cells not expressing RASSF1A, then the authors need to do more work to convince the reader. I.e. many spheroids have to be quantified from both

control cells and RASFF1A expressing cells, and these have to be compared to gels without any cells inside. And please recall that the presence of matrigel also affect collagen bundling.

• Is the orthotopic implantation inducing inflammation in the lung, that could drive collagen remodeling per se. Could they authors i.e. perform a tail-vein injection to prevent inflammation in the lung and recapitulate the data?

Minor Comments:

- All antibodies used in the paper as to be specified with catalog numbers.
- Several places the authors describe the collagen conc. as percentage. Please correct this to mg/ml. There is a huge difference between 3% and 3 mg/ml collagen.
- Some lines are repeated in the M&M under the 'immunoblotting' paragraph
- In the 'Three-dimensional Matrigel migration assay' paragraph: which percentage of Matrigel is use?
- In the 'Immunofluorescence staining in 3D collagen' paragraph: 20% methyl cellulose seems very high. Is this correct? Also, please specific the conc. of TX100 used for permeabilisation during the incubation step.
- In the 'Collagen contraction assay' paragraph: the authors write that the use 8 part collagen-1. Could they specific the concentration of collagen-1 instead?
 - o The authors state that an increase in gel diameter was used to calculate the contraction. The authors obviously mean the decrease in gel diameter. Please correct.
 - o Just a helping suggestion: instead of releasing the gel from the plastic well using a needle, it is much easier to pre-coat the wells with 1%BSA, PBS for 1-2 hours and then plate the gels within the well. BSA-coating prevent gel attachment to the plastic
- It is impossible to see the actual gels in the contraction assay in Fig 2B. Please correct.
 - o The quantification of the 24 well size gel contracted is estimated to be between 1-2 mm³. This is nothing, something must be wrong and should be corrected.
- In general, the SHG images are of very poor quality with exception of Fig 4C.
- Fig 2C: they show SHG of the ipsilateral metastasis at 17days but in Fig1 and in the text they state no metastasis is present at day 17. So, why are they showing this images? The authors are not claiming that they look for pre-metastatic niches - or are they? This reviewer does not understand what this images is telling the reader?
- There is extremely little YAP in the nucleus also in the control cells in Fig 2D. It is hard to judge if the images have been taken in the same z-plane or if the YAP staining is actually not in the cytoplasm under or above the nucleus? Poor quality again.
- The authors claim to have quantified YAP nuclear translocation in Fig 2E. But how do they do this - no description is found in the method section? In addition, it would be more useful to quantify the nuclear/cytoplasmic ratio of individual cells.
- The immunofluorescence detection of P4HA2 is very weak in Fig 3C&D. Are they sure the antibody is detecting the protein by IF. Could the authors provide evidence of this staining after P4HA2 depletion, and could they provide a negative control for the P4HA2 staining in general?
- Is the quantification of collagen derived from the primary tumour or a metastasis in Figure 4C?
 - o Why are they showing a single SHG image of a control-metastasis but not a RASFF1A-metastasis? What do they want to tell with that? Either the authors need to show both cell types or they should remove this image.
- In Fig 5E the graph states fibrotic area. Please simply state that you quantified Picrosirius Red are rather than 'fibrosis area'. Be specific.
- Something is wrong with the display of the images in Figure 6B. For instance, column 2 has one 'green-colored' images while the others are blue?
 - o There are two rows of Merge. What is actually merged here? Please specify.
- Again what is merged in Fig 6F? Dapi and Nanog? Please specify. Importantly these images in Fig 6F seem to be out of the plane, as there is not even a DAPI signal in some of the images (forth row). The Fig 6F would also benefit of some translocation quantification (ratio: nuclear/cytoplasmic signal).
- Why do the authors observe a difference in ipsilateral and contralateral SHG area (Fig 2C and EV2C)?
- What is on the x-axis of Fig EV3D?
- In Fig EV3A: 0.6 or 60% of the Nanog signal does not seem to correspond to the stainings - no way that 60% is in the nucleus when looking at the images!?! This quantification has to be re-evaluated. It seems way to high.

- What do the authors mean by stable collagen (first line page 6)?
- Second line page 7. The authors write 1.1 kPa but the graph shows 11 kPa. Please correct.
- Fig 2C. How did they quantify the collagen volume? How did they end up with 2-8 volume%? Should it not be area at least?

1st Revision - authors' response

3rd Feb 2019

*Reviewer Comments in blue**Author Responses in black*

Referee #1:

They carry out RASSF1A overexpression in one lung adenocarcinoma cell line H1299. These cells are injected in the lung to generate primary tumors. RASSF1A overexpression has no consequence on primary tumor formation although 3/6 injected mice had less metastases when compared to parental H1299 control (con).

1) The statistical significance of these results remains unclear.

While these results were significant we apologize for not including p-value for clarity in all cases.

There are many variables associated to a lung orthotopic injection (Pneumothorax? Lung damage? spilling contralaterally).

We apologize for not including descriptive reasoning for selection of the orthotopic model in the original submission. We chose orthotopic lung model for its accurate recapitulation of the natural tumor environment that allows metastatic dissemination from the lung primary tumor site rather than an ectopic subcutaneous tumor-bearing mouse model. Lung injection was provided by intrathoracic cell injection to avoid pneumothorax and another mechanical damage and has been previously validated (Boehle A.S *et al.* 2000. *Ann Thor. Surg.*; Onn *et al.* 2003. *Clin. Cancer Res*; Servais E. *et al.* 2015. *Curr Protoc Pharmacol.*). Appropriate reasoning for the model selection is now included in the revised version.

Also unclear is whether this is an intrinsic slightly reduced metastatic potential of the H1299 clone selected to carry RASSF1A.

We used a pool of clones to avoid selection effects from a single clone.

So the main result shown in Fig.1 is not impressive (in magnitude) and technically questionable. I would be more convinced if this were complemented by inactivation of RASSF1A in another cell lines (such as the HOP92 used in Fig3) that express it and show that this is increasing metastatic colonization in the same assay.

We also apologise that the original graph omitted 2 data points which contributed to reduced confidence in the results, as also noticed by Rev2. We include these points and as suggested have repeated the orthotopic injections with H1299 and H1299^{RASSF1A} cell lines to further ensure the validity of these findings. The second experiment was also performed from a different pool of clones to further prevent selection bias, shown as differential shading in Fig1 D, E. Moreover, we include additional an in vivo experiment using HOP92 cell line with RASSF1A KO as suggested to further support our data, although technically challenging, as HOP92 been reported to not form orthotopic tumours, we can see tumours with HOP92 stably expressing shRASS1A, Fig EV1E.

2) In Fig2 they provide an intriguing result. They find that RASSF1A expressing cells display less nuclear YAP in cells at an intermediate rigidity. This can have many explanations, but the simplest one is RASSF1A/Hippo signaling may be potentially downstream of - or in a negative feedback with - the cell's response to the ECM. So raising RASSF1A would be expected to blunt YAP by intercepting its activation downstream of the ECM. Instead, and this is the unexpected and interesting finding, they show (pity that this explained imperfectly in the text...) that a different story is going on: the main culprit is not a signal transduction issue within the cell, rather it is the rigidity (or whatever "quality" of the ECM secreted by control cells, but see point below, and not from RASSF1A expressing cells) the dominant signal or main culprit able to trigger nuclear YAP independently of

RASSF1A signaling levels. It seems that ECM mechanics overrules raised RASSF1A signaling, disabling its YAP inhibition, and sending YAP to the nucleus anyway. I think that this section should be extensively rewritten and properly discussed to make this understandable by the non-expert readership.

We thank reviewer and completely see where we introduced the confusion and could have explained this better. We apologize for not discussing our findings in detail and have extensively rewritten the text. We also appreciate the comments of the reviewer in pointing out a potential feedback of ECM through RASSF1A as H1299^{RASSF1A} can achieve nuclear YAP on stiff-ECM, and have made reference to this observation in the revised version.

3) There is however some confusion in these experiments that needs to be addressed. How do they know that H1299 control produce stiffer matrix than the Rassf1 overexpressing one. This is not shown by the second harmonics in the lung (the source of collagen there can be CAFs), and the gel contraction speaks in favor of the differential ability of the two cells to build up a contractile response (that is more related to integrin and cytoskeleton) than to their own ECM production, at least in principle. This aspect needs some dissection and clarification.

We thank the reviewer for these comments and apologize again for not describing this in a clearer manner. Analysis of lung primary tumors or spheroids *in vitro* by SHG only demonstrates organization of collagen I fibres (i.e. we agree this is not stiffness) (original Fig 4A, B, C: **new Fig 4F,G**). We originally provided the stiffness measurement of primary tumors generated by atomic force microscopy (AFM)(original Fig. 5D: **new Fig 4C,D**) to demonstrate the higher deposition of ECM in controls, measured by SHG, correlated with stiffer ECM, e.g. 16kPa compared to RASSF1A 11kPa. We also confirmed this *in vitro*, using H1299 controls and H1299^{RASSF1A} cells cultivated in 3D collagen gels and again, analysis by AFM revealed stiffer ECM produced by H1299 controls also correlated with a denser collagen network SHG. To help clarify we have now reorganised the manuscript with these comments in mind to provide a clearer flow and only present stiff-ECM on actually measuring stiffness with AFM.

We also agree with the reviewer that CAFs are an important part of tumor microenvironment *in vivo* and could be an additional source of collagen production. However they were absent in our *in vitro* analyses suggesting the mechanism of collagen production we present is tumor cell intrinsic. Though, this would make an excellent follow-on question from this study.

An important point raised by reviewer 1 is that that collagen contraction is a property of contractile cells: *'the two cells to build up a contractile response (that is more related to integrin and cytoskeleton'*, however the ECM remodelling and its alignment during cell invasion and tumor progression also correlates with reorganization and contraction of collagen plugs (Mirron-Mendoza M. et al. 2008 MBoC; Gehler S. et al. 2013 Crit Rev Eukaryot Gene Expr., Han et al. 2016 PNAS). We apologize for not appropriately describing our reasoning of this assay and now clarify and discuss this experiment more appropriately (with references) in the revised version.

One way is to show relevance of the P4HA2 gene that is differentially expressed (in the control and rassf1 condition). See next point. For this concern, I would like to see first demonstrated that P4HA2 is essential for gel contraction of H1299 control cells (and thus it is the endogenously produced collagen deposited on top of the synthetic/exogenous/experimental collagen that is responsible for the phenotype).

As requested we include this data (Fig EV2D) indicating expression of P4HA2 mRNA between H1299 control and RASSF1A overexpressing cell lines is different. We also thank reviewer for the experimental suggestion and now provide experiments showing the effect of siRNA targeting P4HA2 and the inhibitor DPCA to further support the collagen contraction experiment and how it influences endogenously produced collagen levels in 2D and 3D by immunofluorescence and SHG (Fig 2G, 3D, 4F, EV2C).

4) The authors are not addressing further the mechanism outlined above in fig2, but rather ask the complementary question of what RASSF1A expression does to the ECM secreted by H1299 cells that disables the positive ECM-YAP axis of control cells. They find reduced P4HA2 (potentially relevant for procollagen folding), although most of their observations on this protein in Figure 5 offers an interesting set of data, but mainly correlative, lacking functional validation. P4HA2 is a central hinge for this paper, hinting to a ECM-YAP-ECM positive feedback loop, but without causality it is hard to appreciate most of these claims. The use of a P4HA2 inhibitory drug of unclear specificity in Figure 6 is not sufficient and need itself validation. Is depletion of P4HA2 sufficient to make the ECM of control cells similar to that one of RASSF1A cells?

We thank reviewer for comments and now provide experiments to validate P4HA2 inhibitor 1,4-DPCA and additional knock-downs using siP4HA2 to demonstrate the effects on ECM we observe are indeed P4HA2 dependent.

5) If they validate the P4HA2 inhibitor 1,4-DPCA, then the finding that reducing YAP-induced ECM stiffness, that in turn sustains/feedbacks on YAP-induced stemness, would start to make much more sense.

We agree that this approach would make the story easier to interpret and have restructured the manuscript accordingly.

6) In this light, the paragraph "Activation of Hippo pathway leads to cancer cell differentiation" really left me struggling. This is at the end of a paper showing that it is the ECM and its properties to be ultimately the determinant of YAP activation and that RASSF1A is a mean to attenuate a positive feedback loop of YAP on the ECM. This should be titled "P4HA2-mediated collagen synthesis attenuates cancer cell differentiation"

We agree and apologize for this misleading title. This is now corrected it to make it more suitable for this section.

7) In figure 7 they use established markers for lung adenocarcinoma progression studies that is mucin and TTF1. However, in previous figures they use Nanog positivity as immediate read-out of YAP activity and ECM rigidity. how established is that?

We apologize for any confusion due to inadequate explanation in the manuscript. OCT4/SOX2 and NANOG are the pluripotency transcription factor cassette in stem cells that maintain pluripotency and numerous studies have provided basic research and clinical evidence for the association with the appearance of cancer stem cells with some providing a direct link from NANOG (e.g. Lin *et al.* 2005. Nat Cell Biol; Liu *et al.* 2017. Molecular Cell). YAP similarly promotes stemness in both ESC and cancer stem cells. We have recently demonstrated a functional link for RASSF1A to YAP regulation and levels of OCT4 and NANOG in embryonic stem cells (ESC) and induced pluripotent stem cells (Papaspuroopoulos *et al.* 2017 Nat Comms). Here we have used NANOG to indicate the appearance of cancer cells that express ESC markers, indicating cancer stem like behaviour. We have clarified the background supporting literature for NANOG as a read out for stemness and hippo signalling in the revised version.

Our hypothesis is that the presence of RASSF1A is associated with a greater degree of lung cancer cell differentiation. This is based on the fact that RASSF1A promotes ESC differentiation (Papaspuroopoulos *et al.* 2018 Nat Comms) and associates with more differentiated lung cancers. We utilised TTF-1 and mucin5B as well characterized markers of terminal lung differentiation in the literature, and linked to better prognosis in lung adenocarcinoma patients, to demonstrate that RASSF1A levels associate with better levels of tumor differentiation. Appropriate referencing is now included to substantiated these approaches.

8) the claim on Wnt signaling is a dramatic detour that goes nowhere and should be deleted. " As expected, ... We interpret this data, together with the isolation of 5T4/TPBG in H1299con ECM to suggest that reduced binding of 5T4/TPBG to the ECM upon P4HA2i allows 5T4/TPBG to suppress WNT signaling and destabilize b-catenin" This is correlative and premature and I have no way to know where is this coming from. Alternative there should be some experimental validation in support of a role of Wnt signaling (note that intrinsic suppression is really hard to follow)

We thank reviewer for comments and apologise for confusion. We agree that our 5T4 data was incomplete, and thus these data are now excluded from the revised version.

9) More generally they are not citing key and supporting references in the result or discussion section that makes the reading very fragmented (I kept asking myself: have they shown that? And where...are they relying on some other data published by others? this is impossible to follow unless one is a really super-expert).

We have taken these points on board and have added appropriate references into the revised manuscript.

10) the title of the paper is misleading. There is very little Hippo signaling investigated here and the connections of RASSF1A with hippo kinases is not thoroughly established in lung tumorigenesis. Thus, a more appropriate title should be RASSF1A controls Tissue Stiffness and Cancer Stem-like Cells in Lung Adenocarcinoma.

We now present a more suitable title of the manuscript that is appropriate for the presented data. In addition, we demonstrate activation status of hippo pathway activation (e.g. pLATS, pYAP) in line with previous published effects of RASSF1A and the association of pathway activity with YAP nuclear localisation and transcriptional activity (Fig. 1B, EV1D).

In sum, the MS is interesting but with important gaps. They should remove and make sure, and restructure the Figure in more logical manner, rather than add data on different directions. Their interpretational lines are often questionable and should be streamlined.

We have substantially rewritten the manuscript based on these valid concerns which we believe has strengthened and simplified the story we are presenting.

Referee #3:

In the paper by Pankova et al, the authors describe a novel mechanism by which loss of RASSF1A induce YAP signaling and induction of tissue stiffness through upregulation of the P4HA2 collagen modifying enzyme. The authors suggest that loss of RASSF1A, a tumour suppressor, and the consequent loss of hippo signaling induces stiffness-dependent beta-catenin signaling that drives cancer stemness and metastatic progression of lung adenocarcinomas but not squamous carcinomas.

The story is interesting, and potentially true, but the quality of the data are often of very low standard and it is difficult to interpret and conclude much of their data. Most of the imaging are of very poor quality and not presenting themselves from a convincing side. The text is poorly written and many typos and mistakes are presented in the text as well as in the figure legends. In general, the data are presented in a confusing manner and no flow is obtained when reading the paper. This reviewer strongly suggests the authors of re-visit the order of the presented data in order to generate a nice flow. The paper in its current form is not of sufficient quality to be published in EMBO Journal. However, with a substantial revision, the study might still reach scientific interest for the readers of EMBO Journal. This reviewer would give the authors the benefit of the doubt if a substantial effort is performed.

Major Comments:

- *A much better characterisation of the cells after overexpression of RASSF1A. Could the authors please show expression levels of the following: pMST1/2, pLATS1/2, pS127-YAP, mRNA levels of a few YAP-target genes and cell proliferation data/curves.*

We agree and have addressed these points in the revised manuscript by including additional pathway activation blots for LATS1 and YAP in H1299 and HOP92 cells (Fig 1B, EV1D) and RT-PCR of P4HA2, CTGF and CYR61 (Fig 3F, EV2D). Additionally, rezasurin based proliferation assays now show that the H1299 isogenic cell lines do not differ in their intrinsic proliferation rates (Fig. 1B) but we do see an increased proliferation rate in HOP92 in the absence of RASSF1A (Fig EV1D), where YAP-TEAD mediated transcription would be expected to be increased. We have included an additional experiment using siRASSF1A in HeLa cells, where endogenous RASSF1A is expressed and hippo pathway active (Matallanas *et al* Mol Cell 2007, Pefani *et al.* EMBO 2018) to further confirm effect on Collagen I to be P4HA2 dependent.

- *The data in Figure 1E representing the metastatic disease are very confusing and surprising for several reasons. First, why are there all 6 control mice having contralateral metastasis but only 5 of the control mice actually have a detectable primary tumours (Fig 1D)?*

We apologise for the lack of clarity, we used MRI for lung primary tumour detection. However, we were not able to detect primary tumour in mouse 2 (M2). One explanation for this event can be, that MRI is able to detect tumours bigger than 5 mm size and thus it was below detection (despite dissemination). However, we could clearly see individual surface tumour nodules on left and right lungs after chest was opened.

- *Second, how come only 4 of the 6 control mice have local ipsilateral mets but all 6 have distant contralateral mets? Please try to explain why some of the mice do not have local mets but only distant mets.*

The presented graph was actually wrong and data points from two mice were not included. We apologise for this error and a corrected version is now included along with an additional in vivo experiment to substantiate these effects.

- *Finally, how was the number of metastasis actually determined/quantified? Could there be some problems with the method?*

We apologise for not making it clear in the description of methods. We counted and quantified lung surface metastatic foci/nodules on day 30 after mice were euthanized. (*as per* Chen Y. *at al.* Journal of Clin. Invest., 2015; Tan X. *et al.* Journal of Clin. Invest., 2017). These description is now included in the text for clarity.

- *The in vitro transwell assays of cell invasion should also be conducted through collagen-1. First of all, collagen seem to be a far more important contributor to the molecular mechanism presented in the study (although Lamin B2 is upregulated in their mass spec data). Second of all, invasion*

through matrigel mimics invasion through basement membranes while invasion through collagen mimic invasion through the parenchyma tissue.

We fully agree with reviewer that invasion through collagen-1 is more complex and mimics invasion through the parenchyma tissue. However, as lung tissue is highly enriched with laminins, the most abundant structural non-collagenous glycoproteins of basement membrane, we have used matrigel matrix (80% -laminin) for human H1299 invasion to mimic a normal lung microenvironment. As requested we now provide collagen -1 invasion (Fig. 3A) and have included this important point in the text.

• One major issue through the paper is whether tissue stiffness or the actual collagen concentration is to be responsible for their observations. When using higher conc. of collagen it is true that the stiffness increases but also the available epitopes (avidity) for cell binding. It is therefore very important to validate what is driving the progression in their study. This reviewer demand to see experimental set-ups, which discriminate between stiffness and collagen concentration. One way of validating stiffness vs collagen conc. is to use polyacrylamid gels of different stiffness and then coat them with the same collagen conc. One can even purchase custom made gels from Matrigen (see softwells). This reviewer is in doubt if it is actually stiffness and not collagen avidity that drives Nanog expression. Indeed, Fig EV3D clearly demonstrate that cell plated on 2D glass, which is much stiffer than 3% (3 mg/ml) collagen actually have lower levels of Nanog in the nucleus!!!!

We thank reviewer for these comments and suggestions; we fully agree and think that both tissue stiffness and collagen concentration contributes to our observations. We apologise that this was not satisfactorily addressed. To further define and clarify, we now provide additional experiments with custom made gels for stiffness and collagen concentration to validate NANOG expression (Fig 5A,C EV4A). Interestingly, we found that 4kPa was in agreement with our previous results but that very-stiff (25kPa) matrix appeared to be inhibitory and in keeping with the 2D glass. We provide a discussion point on this to potentially explain this phenomenon suggesting that very stiff-ECM may be too rigid to expose binding sites, supported by the literature.

• A second major problem with this paper is that the mechanism is proposed to go through P4HA2. But only one experiment using a P4HA2 inhibitor is used to rescue the effect of Nanog expression. No validation of the inhibitors specificity is shown, and no other rescue experiments have been conducted. This reviewer demands that rescue experiments are conducted using ablation of P4HA2 in cells lacking RASSF1A expression. These should include, gel contraction, collage production, ECM stiffness, beta-catenin, stemness (Nanog, Oct4, SOX expression) and potentially also include in vivo validation (although the in vivo part may seem harsh to demand).

We agree with reviewer that these experiments would be definitive and we now include all mentioned experiments by reviewer with siRNA against P4HA2, to verify our mechanism, however as suggested *in vivo* data was too difficult to accumulate in a reasonable time, as pointed out.

• The experimental data on 5T4 proteins are extremely weak. I think the data does not allow any kind of conclusions without further experimentation.

We agree with reviewer that our 5T4 data is preliminary, and we now exclude this data to support a clearer story.

• In the model in Figure 8, the author mention OCT4 and SOX2, but no data is presented in the paper! Could they include data on these two genes as well as Nanog?

We again apologize, the model was meant to represent the pluripotency cassette through which NANOG is expressed in an OCT4 dependent manner. Now, we include levels of mRNA of these additional genes to support the stemness phenotype (Fig 6A,D) and supported by additional western blot data for protein level (Fig 6C, F) .

• The authors show that control cells produce linearized collagen bundles emanating from the spheroid in Fig 4A. As the spheroids are made within collagen matrices this is not so surprising. In fact, one should even detect collagen bundles like that in gels without any cells. So, if the conclusion is that more bundles are produced or formed by the cells not expressing RASSF1A, then the authors need to do more work to convince the reader. I.e. many spheroids have to be quantified from both

control cells and RASFF1A expressing cells, and these have to be compared to gels without any cells inside. And please recall that the presence of matrigel also affect collagen bundling.

We have used hanging drop method for spheroid generation which are embedded into collagen matrix after their aggregation. For collagen gels, we have used 2mg/ml concentration of non-cross-linked rat tail collagen I gel, that should be not detected by SHG, as this only allows the detection of native, self-assembly, polarized collagen fibres with non-centrosymmetrical molecular structure. Thus, we believe that collagen bundles emanating from spheroids are produced by cells within the spheroids and not artificially by collagen gel. We apologise for causing any confusion or misrepresentation regarding bundles due to our failure to appropriate discuss these points properly in the original text. In short, our observations revealed that H1299 cells produce more collagen (not bundles) and the cells are able to remodel collagen fibres into highly organized collagen structures (*in vitro* and *in vivo*) which allows single cell invasion from primary spheroid (*as per* Han W. *et al.* 2016 PNAS). We have revised the text to ensure this point is clear and apologise for the lack of clarity in the original manuscript.

- *Is the orthotopic implantation inducing inflammation in the lung, that could drive collagen remodeling per se. Could they authors i.e. perform a tail-vein injection to prevent inflammation in the lung and recapitulate the data?*

We thank reviewer for this suggestion. We agree, that inflammation could drive collagen remodeling and composition by recruiting immune cells and the secretion of various inflammatory cytokines. We stained lung tumor primary tissue for macrophages and did not see major differences in H1299^{control} tumors compared to those expressing RASSF1A. We also failed to appropriately discuss the reason for using the orthotopic model. As also outlined for a similar concern of reviewer 1, we apologize for not including descriptive reasoning for selection of the orthotopic model in the original submission. We chose the orthotopic lung model for its accurate recapitulation of the natural tumor environment that allows metastatic dissemination from the lung primary tumor site rather than an ectopic subcutaneous tumor-bearing mouse model. Lung injection was provided by intrathoracic cell injection to avoid pneumothorax and another mechanical damage and has been previously validated (Boehle A.S *et al.* 2000. *Ann Thor. Surg.*; Onn *et al.* 2003. *Clin. Cancer Res*; Servais E. *et al.* 2015. *Curr Protoc Pharmacol.*). This more appropriate reasoning for the model selection is now included. We feel that while tail vein injection could address the metastatic potential of individual cells, it would not address the tumor microenvironment of a primary tumor. We now include further *in vivo* experiments and show macrophage staining to demonstrate the equivalency of the isogenic cell tumors in this regard (Fig EV1C).

Minor Comments:

- *All antibodies used in the paper as to be specified with catalog numbers.* Now, included.
- *Several places the authors describe the collagen conc. as percentage. Please correct this to mg/ml.*
There is a huge difference between 3% and 3 mg/ml collagen. corrected for % for mg/ml.
- *Some lines are repeated in the M&M under the 'immunoblotting' paragraph.* Corrected.
- *In the 'Three-dimensional Matrigel migration assay' paragraph: which percentage of Matrigel is use? We used coating of pure Matrigel matrix at a concentration 8mg/ml.*
- *In the 'Immunofluorescence staining in 3D collagen' paragraph: 20% methyl cellulose seems very high. Is this correct? Also, please specific the conc. of TX100 used for permeabilisation during the incubation step.* 20% methylcellulose is correct, concentration of TX100 is 0.3% for permeabilization.
- *In the 'Collagen contraction assay' paragraph: the authors write that the use 8 part collagen-1. Could they specific the concentration of collagen-1 instead?* Corrected to final concentration of collagen of 2.5 mg/ml.

o *The authors state that an increase in gel diameter was used to calculate the contraction. The authors obviously mean the decrease in gel diameter. Please correct.* Corrected.

o Just a helping suggestion: instead of releasing the gel from the plastic well using a needle, it is much easier to pre-coat the wells with 1%BSA, PBS for 1-2 hours and then plate the gels within the well. BSA-coating prevent gel attachment to the plastic. We thank very much reviewer for suggestion! We have modified our protocol.

- *It is impossible to see the actual gels in the contraction assay in Fig 2B. Please correct.* Better pictures of gel contraction are now included with an indication of the original well size to illustrate the degree of contraction being measured.

o The quantification of the 24 well size gel contracted is estimated to be between 1-2 mm3. This is nothing, something must be wrong and should be corrected. Corrected.

- *In general, the SHG images are of very poor quality with exception of Fig 4C.*

We have attempted to show better images or zoom images to increase confidence in SHG.

- *Fig 2C: they show SHG of the ipsilateral metastasis at 17days but in Fig1 and in the text they state no metastasis is present at day 17. So, why are they showing this images? The authors are not claiming that they look for pre-metastatic niches - or are they? This reviewer does not understand what this images is telling the reader?*

We apologise for this error, the SHG image shows primary tumor on day 17.

- *There is extremely little YAP in the nucleus also in the control cells in Fig 2D. It is hard to judge if the images have been taken in the same z-plane or if the YAP staining is actually not in the cytoplasm under or above the nucleus? Poor quality again.*

Improved images have been included and great level of validation of nuclear YAP is included throughout.

- *The authors claim to have quantified YAP nuclear translocation in Fig 2E. But how do they do this - no description is found in the method section? In addition, it would be more useful to quantify the nuclear/cytoplasmic ratio of individual cells.*

We apologise to reviewer, quantification was done by Fiji-colocalization software as a nuclear/cytoplasmic ratio and is outlined in methods.

- *The immunofluorescence detection of P4HA2 is very weak in Fig 3C&D. Are they sure the antibody is detecting the protein by IF. Could the authors provide evidence of this staining after P4HA2 depletion, and could they provide a negative control for the P4HA2 staining in general?*

We now provide P4HA2 staining after its depletion as well as negative control.

- *Is the quantification of collagen derived from the primary tumour or a metastasis in Figure 4C?*

We apologise to reviewer for misunderstanding, its collagen quantification from primary tumors.

o Why are they showing a single SHG image of a control-metastasis but not a RASSF1A-metastasis? What do they want to tell with that? Either the authors need to show both cell types or they should remove this image. We thank reviewer for comments, we wanted to show that micrometastasis generated by H1299 control cells also produce collagen fibres around the microtumor in the same pattern as we observed *in vitro* around spheroids embedded in the collagen gels. We have not included RASSF1A as we could not see any collagen fibres in metastases. We now include the additional image (Fig 4G).

- *In Fig 5E the graph states fibrotic area. Please simply state that you quantified Picrosirius Red are rather than 'fibrosis area'. Be specific.*

We have corrected this as quantification of picrosirius red staining.

- *Something is wrong with the display of the images in Figure 6B. For instance, column 2 has one 'green-colored' images while the others are blue? There are two rows of Merge. What is actually merged here? Please specify.*

We apologise to reviewer for this confusion, merges are with DAPI. Green-colored image is due to high levels of Nanog in control cells make the merge with DAPI appear very bright green. The figures now refer to NANOG:DAPI correlation to indicate exactly what is being measured.

- *Again what is merged in Fig 6F? Dapi and Nanog? Please specify. Importantly these images in Fig 6F seem to be out of the plane, as there is not even a DAPI signal in some of the images (forth row). The Fig 6F would also benefit of some translocation quantification (ratio: nuclear/cytoplasmic signal).*

We apologise to reviewer for poor quality of images, caused by technical issues. The merge represents DAPI and Nanog staining.

- *Why do the authors observe a difference in ipsilateral and contralateral SHG area (Fig 2C and EV2C)?* Original Fig 2C and EV2C demonstrated different composition of collagen fibres between primary H1299 control tumors in ipsilateral lungs and unorganized and fused in non-metastatic contralateral lungs (as a control).
- *What is on the x-axis of Fig EV3D?* We apologize reviewer for misunderstanding, each dot on x-axis represents analysed cells for experiments (20 cells/per dot).
- *In Fig EV3A: 0.6 or 60% of the Nanog signal does not seem to correspond to the staining - no way that 60% is in the nucleus when looking at the images!? This quantification has to be re-evaluated. It seems way to high.*

We apologize reviewer for not making this clear. This is analysis based on Pearson colocalization coefficient 0.6 that means Nanog colocalization with DAPI in the nucleus, not signal intensity in the nucleus. The closer is Pearson coefficient to value 1, the more colocalization is involved.

- *What do the authors mean by stable collagen (first line page 6)?*

We were referring to stable collagen fibres and have reworded to ensure clarity.

- *Second line page 7. The authors write 1.1 kPa but the graph shows 11 kPa. Please correct.* We apologize for this error, it should be 11 KPa.

- *Fig 2C. How did they quantify the collagen volume? How did they end up with 2-8 volume%? Should it not be area at least?*

Analyses was provided by Fiji software to calculate collagen area and we agree it should be area rather than volume.

2nd Editorial Decision

21st Mar 2019

Thank you for submitting your revised manuscript for consideration by The EMBO Journal, and your patience with our response. My sincere apologies for the delay in processing your manuscript, which was due to much delayed referee input. Your revised study was sent back to the two original referees for re-evaluation. Please find their comments enclosed below.

As you will see, referee #3 remains overall more critical on the study, however we decided - in light of the strong support of the other referee - to give you the opportunity to revise your manuscript to address the referee's points.

In more detail, referee #1 finds that his/her concerns have been sufficiently addressed and is now broadly in favour of publication. However, while referee #3 agrees that the link between RASSF1A, P4HA2 and collagen deposition is now more convincing, this referee remains critical regarding the ECM stiffening assay data added, pointing to concerns regarding experimental design and missing controls. In addition, referee #3 has persistent reservations regarding the claims made on upstream control of cancer stem-like transcription factors and tumor dedifferentiation by RASSF1A-dependent ECM stiffness, and points to insufficient data quality of the data, which in his/her view undermines the robustness and impact of the latter results. Further, this referee asks you to improve the overall presentation of the results and complement the methods annotation.

While we usually only offer one single round of revision at The EMBO Journal, considering the positive comments of referee #1, we have decided to ask you to revise your manuscript regarding the points raised by referee #3. As this would obviously require a substantial amount of additional work, you might however alternatively consider transfer of your work to our sister EMBO Reports and reworking the manuscript into a shortened version in order to publish the study in a reasonable time frame. As to your preference at submission, I have enquired back and discussed the work with my colleague Achim Breiling, who would be positive about the latter scenario.

While we leave above choice entirely up to you, I would appreciate if you could contact me during the next days and let us know if you engage in compelling additional revisions in case I would not close the file for EMBO Journal.

 REFEREE REPORTS:

Referee #1:

I think this is an overall sound paper. The authors addressed to substantial extent my concerns. The paper does offer new insights connecting an important tumor suppressor to the control of the microenvironment and in particular the mechanical microenvironment, as such impacting on cell states

Referee #3:

In the revised paper by Pankova et al, the authors have provided more data on their model system (H1299 cells) and most importantly the in vivo data are now more convincing. The data in Fig 1 now convincingly show that RASSF1A expression correlate with less metastasis in lung cancer. The authors also show convincingly, that RASSF1A regulate P4HA2 and collagen production (Fig 2) and that this regulate ECM remodeling and the ability of H1299 cells to invade collagen/matrigel in vitro (Fig 3). The authors then demonstrate that these effects are dependent on YAP1 signaling, which ultimately generates a positive feed forward loop to promote more ECM remodeling and further YAP translocation into the nucleus (Fig 3G).

The authors then show that RASSF1A suppresses ECM stiffening in vitro and in lung cancer tissue using AFM and staining for fibrillar collagen (picosirius red). These data seem convincing but they then try to underline their findings by performing SHG imaging of lung tissue as well as performing various in vitro assays using collagen gels. These experiments are far from convincingly performed and the data are not convincing either. This reviewer strongly suggests to remove these data (Fig 4F&G) unless the authors are ready to repeat these experiments in order to provide convincing data?

This reviewer already mentioned some concerns regarding Fig 4F&G in the first revision. I hereby repeat some of my opinions:

In Figure 4F, the authors still do not show SHG of empty gels without cells. These gels should still have SHG signal if the collagen-gel is well polymerized under appropriate pH. If the authors have problems with the SHG they could try to image the fibrils using reflectance microscopy (confocal). The authors also claim they use non-cross-linked rat-tail collagen. Normally rat tail collagen is extracted using acid that maintain telo-peptides and cross-links. If their collagen on the other hand has been extracted using pepsin-digestion, which cleaves the telo-peptides - then they have no cross-links. Please be sure whether your collagen is cross-linked or not. The quantification in Fig 4F shows that treatment of DPCA gives the lowest SHG signal, but that is clearly not the case when looking at the image. Here DPCA is still showing nice linearized collagen (top-left corner). This reviewer believes that quality of these experiments are still too poor to be used in a publication. Obviously, had the images been of higher resolution it might have been possible to evaluate properly...but the provided images are of such low quality that no conclusions should be made. Healthy lung tissue has plenty of SHG signal and the topology is very defined. Did the authors really scrutinize the difference in SHG structures between healthy lung and c H1299 and H1299RASSF1A tumours? A better and more comprehensive comparison would be appreciated. In the revised version, the authors now start speaking about pre-metastatic niches without demonstrating that this is what they have. How can they exclude small micro-metastasis at Day 17 without doing some HE staining's of the tissue? The authors then claim to measure differences in SHG in these pre-metastatic niche. In theory this could be true, but their data does not support these conclusions. In fact, this reviewer cannot appreciate any differences in the SHG images provided in fig EV3. The p-value in the graph is not visible either.

Until Figure 4/5 the paper has been improved and the flow is decent. But from figure 5 the authors

now want to enter into the Cancer stem cell world and how their findings might affect stemness genes like Nanog, Oct4, SOX2 and CD133 and the differentiation state of lung cancers. Not only are their data not convincingly presented as all immunofluorescence images are of such low quality that it is impossible to evaluate. Their data jump between various genes and stemness markers and between various in vitro assays without showing anything properly. No in-depth information is provided - just fractions of scattered data. The reader gets the feeling of being thrown in different directions without getting any final answers. The interesting observation of Nanog being tightly regulated by the stiffness obviously does not make it more easy to understand. This reviewer feels that more mechanistic understanding is needed or maybe the authors should focus on a few observations instead of spreading out to thin. Some confusing observations: In matter of fact, in Fig 6B (IF), all Nanog is gone after siYAP. This is not the case looking at WB in Fig 6C&F. In Fig 6F, the loss of P4HA2 by siRNA does not really decrease the SOX2, Nanog and Oct4. Only the treatment of the inhibitor does -why?

The authors then decide to bring even more confusion and uncertainties in Figure 7 by introducing various markers of differentiation. Yes, this is interesting but again no explanation of the findings is provided. Why do they show Fig 7D with cells plated on matrigel? How can these data help the story? This reviewer suggests to delete all panels except Fig 7B, which tells everything in one simple figure.

In summary, the story is still interesting, but unfortunately the resolution of the most figures is simply too poor to be accepted for publication. For instance, most panels and sub-figures including text are totally impossible to read (even in the tiff-files). How can we as reviewers evaluate such poor quality figures? Try to evaluate the IHC staining's in Fig 2C, or try to read the p-values in Fig 2G, or even evaluating the transwell images in Fig 3A-C!!! Pretty much half the panels of all figures are of too low resolution and impossible to interpret. Indeed, this reviewer had to trust the text in order to understand the paper! This is not acceptable for a revised paper.

Although the text is improved compared to the first revision, the paper still loses its coherence from figure 4 and onwards. The authors try to put too many things into the papers which are very poorly depicted, and instead of making the story more interesting, it makes it more confusing and at the end gives an untrusted feeling. This is a pity as the overall story is interesting!

This reviewer gave the authors the benefit of the doubt after the first revision. But the revised paper has not convinced this reviewer that the paper deserves publication in EMBO Journal.

Minor comments:

Page 3, line 6: YAP/TAZ are not transcription factors, they are transcriptional co-activators.

In Fig 3D. There is still a problem with the quantification. How can you say that the gels have contracted less than 1 mm²? The siNT has clearly contracted more than 0.8 mm².

The authors state that their 3D spheroids are app. 300 nm in diameter. They are much bigger and they probably meant 300 um. Please correct.

Fig EV4C. How can the authors be sure that their anti-HIF antibody works without a positive control like low oxygen or treatment with DMOG or CoCl₂?

2nd Revision - authors' response

25th March 2019

Many thanks for giving us the opportunity to respond. Indeed we immediately see where the majority of the problem is, the figure resolution is appalling in the revised figures. We had ensured everything was correct in the *.pptx files, but in conversion to TIFF and PDF the resolution completely dropped. I have attached a PDF of Figure 4F,G and EV3 (including zooms) to illustrate the exact point the reviewer had with interpretation, and they were completely correct.

This is readily fixable as we have much better quality images (in fact a lot of these were present in the original submission with good quality). I have also put together a point by point response to the reviewers specific points and have responded to the image and additional issues.

Would it be OK to discuss by phone how to proceed? Again, I apologise for this error it should not have happened.

Paste in PDF

3rd Editorial Decision

11th Apr 2019

Thank you for following-up on our decision and sending the complementary high-resolution figures, which clearly helped to resolve the issues raised by the referee. Thank you also for your patience with our response.

I have now shared the new material with referee #3, in light of which this reviewer is much more positive now about the work. Please see his-her additional comments enclosed below.

Together with the support of referee #1, I thus encourage you to do a final minor revision of the study, introducing additional discussion points and caveats where appropriate.

There are also a number of formatting issues, which need to be addressed at re-submission. Please see the list enclosed below.

REFEREE REPORTS:

Referee #3, additional comments:

I went through the new figures and the rebuttal. Now, being able to see the figures and I am more satisfied. I think the paper is looking fine now.

However, I am still not impressed by any of their SHG images. But the data are relatively supported by other evidence.

For instance try to compare their SHG imaging in EV3 with Figure 6 and Sup Fig 2 in the paper 'ISDoT: in situ decellularization of tissues for high-resolution imaging and proteomic analysis of native extracellular matrix', Nature Medicine 2017.

3rd Revision - authors' response

23rd Apr 2019

We have now performed the remaining editorial concerns and address the final reviewer's comments below. We hope you know find the manuscript acceptable for publication.

Referee#3, additional comment:

I went through the new figures and the rebuttal. Now, being able to see the figures and I am more satisfied. I think the paper is looking fine now. However, I am still not impressed by any of their SHG images. But the data are relatively supported by other evidence. For instance try to compare their SHG imaging in EV3 with Figure 6 and Sup Fig 2 in the paper 'ISDoT: in situ decellularization of tissues for high-resolution imaging and proteomic analysis of native extracellular matrix', Nature Medicine 2017.

We thank reviewer for his additional comments. We compared our SHG in vivo images (Fig.EV3) and compared with reviewer's recommended article published in Nature Medicine, 2017. Our images do differ from those in this article as they monitored SHG collagen deposition together with immunostaining of collagen of lung macrometastases after implantation of 4T1 breast cancer cells into mammary pads. We employed direct orthotopic injection of H1299 cells for primary tumor bearing in the lungs which were ~0.5mm in size at day of our experiment, while their primary mammary tumours reached 10 mm (20 fold greater) and a likely reason why these authors observed more prominent collagen deposition. Moreover, authors utilised high-resolution microscopy for detection of SHG signals, whereas our images were obtained using a conventional confocal microscope.

Corresponding Author Name: Eric O'Neill

Manuscript Number: EMBOJ-2018-100532R